# Probabilistic Visualization and Analysis of User Preferences via Random Fourier Features

## Abstract

Understanding user preferences plays a crucial role in domains where strategies are designed by domain experts, such as personalized recommendations, targeted marketing, and human-centered interface design. However, many existing methods prioritize predictive accuracy over model transparency, limiting their use in settings that require interpretability. To address this issue, we propose the **Random Fourier Feature Shared Latent Variable Model (RFSLVM)**, a probabilistic generative model that integrates ideas from shared Gaussian process latent variable models and random Fourier feature (RFF) approximations for interpretable user preference analysis. RFSLVM jointly models two data modalities: real-valued item features and binary user ratings. It learns a *two-dimensional* **visualization space** that captures relationships among items and user ratings. Additionally, it infers *user-specific* **preference vectors** (weight vectors in RFF space) that are compact and continuous representations of generally nonlinear preference functions. These vectors support tasks such as measuring user similarity and performing preference-based clustering, thereby facilitating downstream analysis and decision-making. We evaluate RFSLVM on multiple real-world datasets and find that it performs competitively against baseline models, while maintaining interpretability. In addition, we demonstrate the utility of the learned representations through qualitative analyses, including hierarchical clustering and the identification of latent preference patterns. These findings suggest that RFSLVM offers a practical and interpretable approach to modeling user preferences in real-world applications, particularly suitable for expert-in-the-loop analysis.

## 1 Introduction

Understanding user preferences plays a crucial role in domains where strategies are designed by domain experts, such as personalized recommendations (He et al., 2017; Kang & McAuley, 2018; Sun et al., 2019), targeted marketing (Min et al., 2023; Liu et al., 2023), and human-centered interface design (Kunkel et al., 2017; Zhang et al., 2020). Despite recent advances in automation, many strategies in these domains are still crafted manually by domain experts. These expert-designed strategies rely on a clear understanding of user preferences, as reflected in how users engage with items. To inform such strategies, interpretable visualizations (Yuan et al., 2021; Li & Zhao, 2021) and multi-perspective analyses of user preferences provide useful insights.

This study considers scenarios with two data modalities: real-valued item features and binary user ratings indicating positive or negative feedback on the items. Based on these inputs, we address two main objectives. First, we aim to derive a shared latent space, referred to as the **visualization space**, that captures relationships among items and user ratings while supporting intuitive, low-dimensional visualization. Second, we aim to learn vector representations, referred to as the **preference vectors**, that encode users' nonlinear preferences within the visualization space. These vectors support the computation of similarities between users, facilitating exploratory analysis such as clustering and preference pattern discovery. An example of the output produced by our model is shown in Figure 1, which illustrates a hierarchical clustering of user preferences based on learned latent representations. The figure highlights both the cluster structure among

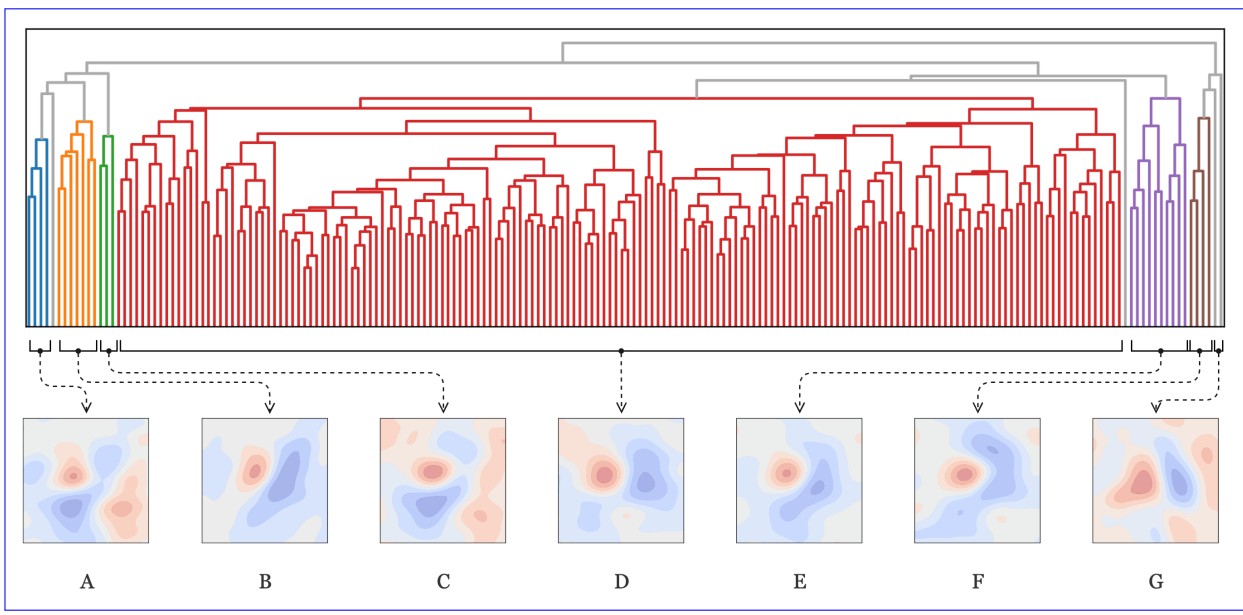

Figure 1: **Hierarchical Clustering of User Preferences**. This figure is constructed based on the visualization space and preference vectors inferred by RFSLVM in Section 5.3. The dendrogram visualizes the hierarchical structure of user preferences. Each leaf node represents an individual user, and branch colors indicate distinct clusters. Contour diagrams show the distribution of user preferences within the visualization space for each cluster. In the contour diagrams, color indicates preference intensity, increasing from blue to red.

users and the characteristic preference patterns associated with each cluster. Figure 2 embeds items in the visualization space and illustrates how preference intensity varies across items, using the preference pattern of Cluster A as an example.

Dimensionality reduction techniques such as Principal Component Analysis (PCA) (Hotelling, 1933), t-distributed Stochastic Neighbor Embedding (t-SNE) (van der Maaten & Hinton, 2008), and Uniform Manifold Approximation and Projection (UMAP) (Huang et al., 2022) are widely used to produce low-dimensional visualizations. Similarly, probabilistic latent variable models such as Gaussian Process Latent Variable Models (GPLVMs) (Lawrence, 2003; Titsias & Lawrence, 2010; Lalchand et al., 2022) and Random Feature Latent Variable Models (RFLVMs) (Gundersen et al., 2021; Zhang et al., 2023) are also used for visualization of nonlinear structure. While these methods offer useful low-dimensional representations, they do not support multiple modalities or provide preference vectors for downstream tasks.

Beyond single-map methods, multiple maps t-SNE (Van der Maaten & Hinton, 2012) represents non-metric similarities using multiple low-dimensional maps, and MR-SNE (Mizutani et al., 2020) extends this idea to multimodal relational data by jointly embedding cross-domain relations and within-domain neighborhoods. Related neighborhood-preserving ideas have also been explored in multi-view or aligned variants of UMAP. These SNE- and UMAP-based approaches are primarily designed for visualization and do not explicitly model user-specific preference vectors.

To model multiple modalities within a probabilistic framework, Shared Gaussian Process Latent Variable Models (SGPLVMs) (Ek, 2009; Salzmann et al., 2010; Lalchand & Eilers, 2025) and Manifold Relevance Determination (MRD) (Damianou et al., 2012; 2021) extend GPLVMs by introducing shared latent spaces. More recent work has explored multi-view extensions based on spectral mixture kernels, improving scalability and expressiveness (Yang et al., 2025). Nevertheless, these models typically assume Gaussian observation models and do not explicitly provide user-specific preference vectors.

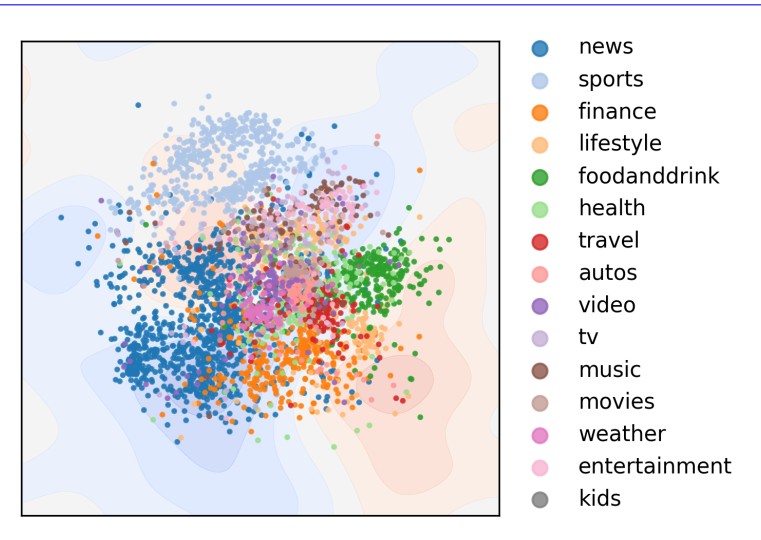

Figure 2: **Interpreting preference patterns in the visualization space.** In the scatter plot, each point represents an item (e.g., a news article) in the visualization space. Point colors indicate item categories defined in the MIND dataset Wu et al. (2020). In the contour plots, color represents preference intensity, increasing from blue (low) to red (high). This visualization shows which news categories are associated with high or low preferences.

Neural architectures such as Multimodal Variational Autoencoders (MMVAEs) (Shi et al., 2019; Suzuki & Matsuo, 2022) also integrate heterogeneous modalities into shared latent spaces using deep encoder–decoder networks. Recent extensions (Palumbo et al., 2023; Märtens & Yau, 2024; Sutter et al., 2024; Qiu et al., 2025) further improve coherence and generative quality by disentangling shared and private latent factors. However, these models often rely on post hoc 2D visualizations (e.g., applying t-SNE to learned representations) and do not explicitly provide preference vectors.

In this study, we adopt a Gaussian process (GP) based approach over neural architectures for two main reasons. First, compared to deep neural networks, GP models offer more transparent, white-box modeling of nonlinear user preferences, which better aligns with our goal of interpretability. Second, prior work suggests that GP-based models can perform well even with limited data Damianou et al. (2021).

We propose the **Random Fourier Feature Shared Latent Variable Model (RFSLVM)**, a probabilistic generative model that integrates ideas from GPLVM, RFLVM, SGPLVM, and MRD. All observations are conditionally generated from a shared, two-dimensional latent space referred to as the **visualization space**, designed to promote interpretability. This space is mapped into a high-dimensional feature space via Random Fourier Features (RFFs) (Rahimi & Recht, 2007), enabling efficient inference and expressive modeling. User preferences are modeled as linear weight vectors, referred to as **preference vectors** (weight vectors in the RFF space), allowing the model to capture complex, nonlinear preference structures. The model also explicitly accounts for heterogeneity between real-valued item features and sparse binary user ratings. Table 1 summarizes the capabilities of RFSLVM relative to previous methods in terms of multimodality, 2D visualization, and preference vectors.

We evaluate RFSLVM on real-world datasets and observe both predictive and reconstruction performance comparable to prior approaches, while preserving interpretability. We further demonstrate practical analyses enabled by the learned visualization space and preference vectors, including user clustering and identification of latent preference patterns, highlighting its suitability for expert-in-the-loop analysis.

The primary contributions of this study are summarized as follows:

| Method Family | Multimodality | 2D Visualization | Preference Vectors |
|---|---|---|---|
| PCA / t-SNE / UMAP | ✗ | ✓ | ✗ |
| GPLVM / RFLVM | ✗ | ✓ | ✗ |
| multiple maps t-SNE / MR-SNE | ✓ | ✓ | ✗ |
| MMVAE | ✓ | ✓[2] | ✗ |
| SGPLVM / MRD | ✓[1] | ✓[2] | ✗ |
| **RFSLVM (Ours)** | ✓ | ✓ | ✓ |

Table 1: **Comparison of RFSLVM with previous methods. Multimodality:** Supports heterogeneous modalities (e.g., real-valued item features and binary user ratings). **2D Visualization:** The latent space is optimized in two dimensions during training, enabling direct visualization without post-processing. **Preference Vectors:** The model provides user preference vectors. ✓: supported; ✗: not explicitly supported. [1]: Assumes Gaussian outputs and does not natively support Bernoulli outputs, requiring extensions to handle binary user ratings. [2]: Shared latent spaces are not restricted to 2D and often require post hoc projections (e.g., t-SNE) for 2D visualization.

1. We propose RFSLVM, a probabilistic generative model that represents multiple data modalities in a shared two-dimensional **visualization space**, enabling interpretable analysis of user preferences through **preference vectors**.

2. We adopt an efficient inference scheme using RFFs to handle heterogeneous data, including real-valued item features and sparse binary user ratings.

3. We present empirical results on real-world datasets, demonstrating competitive predictive and reconstruction performance, and enabling practical analyses such as user clustering and the discovery of latent preference patterns.

The remainder of this paper is organized as follows. Section 2 reviews relevant background and related work, highlighting key challenges that motivate our approach. Section 3 introduces RFSLVM and formulates the inference objective. Section 4 presents a quantitative evaluation of RFSLVM on real-world datasets, reporting predictive performance, reconstruction accuracy, and inference time. Section 5 demonstrates practical applications enabled by the visualization space and preference vectors. Finally, Section 6 concludes the paper and outlines directions for future research.

## 2 Background and Related Work

In this section, we review related work that forms the foundation of our approach. We also identify key challenges that motivate further extensions.

### 2.1 Gaussian Process Latent Variable Models

The Gaussian Process Latent Variable Model (GPLVM) (Lawrence, 2003; Titsias & Lawrence, 2010; Lalchand et al., 2022) is a probabilistic generative model for nonlinear dimensionality reduction that explains high-dimensional data as generated from a low-dimensional latent space via Gaussian processes (GPs).

Let $\mathbf{X} = \{\mathbf{x}_n\}_{n=1}^N \in \mathbb{R}^{N \times D}$ denote the observed features of $N$ items in a $D$-dimensional space. Let $\mathbf{Z} = \{\mathbf{z}_n\}_{n=1}^N \in \mathbb{R}^{N \times Q}$ be the corresponding latent representations, where $Q \ll D$. Each observed dimension is modeled independently using a $\mathcal{GP} : \mathbf{X}_{:,d} \sim \mathcal{N}(\mathbf{m}, \mathbf{K})$. Here, $\mathbf{m}$ is the mean vector, typically set to $\mathbf{0}$, and $\mathbf{K}$ is the kernel (covariance) matrix with entries $K_{i,j} = k(\mathbf{z}_i, \mathbf{z}_j; \boldsymbol{\theta})$ for $i, j \in \{1, \dots, N\}$. The Radial Basis Function (RBF) kernel is commonly used:

$$k_{\mathrm{RBF}}(\mathbf{z}_i, \mathbf{z}_j; \boldsymbol{\theta}) = \theta_1 \exp\left(-\frac{\|\mathbf{z}_i - \mathbf{z}_j\|^2}{2\theta_2}\right), \tag{1}$$

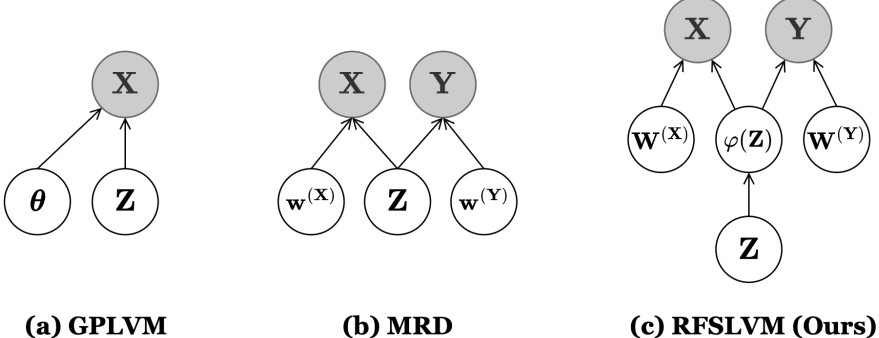

**(a) GPLVM**          **(b) MRD**          **(c) RFSLVM (Ours)**

Figure 3: **Related Models and RFSLVM.** White nodes represent latent variables or model parameters, gray nodes represent observed data (modalities), and arrows indicate probabilistic dependencies. RFSLVM (ours) extends MRD with two key modifications: (i) the latent space $\mathbf{Z}$ is constrained to two dimensions to enhance visual interpretability; (ii) $\mathbf{Z}$ is transformed via RFFs into $\varphi(\mathbf{Z})$, and a linear model is applied to each output dimension of each modality. This enables the modeling of preference vectors $\mathbf{w}_h^{(\mathbf{Y})} \in \mathbf{W}^{(\mathbf{Y})}$.

where $\boldsymbol{\theta} = \{\theta_1, \theta_2\}$ are kernel hyperparameters. The graphical model of GPLVM is shown in Figure 3(a).

GPLVMs have two major limitations for our purpose. First, they assume a single modality, which motivates extensions such as Shared Gaussian Process Latent Variable Models (see Section 2.2). Second, inference scales cubically with data size due to matrix inversion, making it impractical for large datasets. To address this limitation, inducing point methods (Quiñonero-Candela & Rasmussen, 2005; Snelson & Ghahramani, 2005) and random Fourier features (see Section 2.3) have been proposed. Variants known as Random Feature Latent Variable Models (RFLVMs) (Gundersen et al., 2021; Zhang et al., 2023) employ kernel approximations via RFFs. However, like GPLVMs, they also remain limited to a single modality.

## 2.2 Shared Gaussian Process Latent Variable Models

The Shared Gaussian Process Latent Variable Model (SGPLVM) (Ek, 2009; Salzmann et al., 2010) and Manifold Relevance Determination (MRD) (Damianou et al., 2012; 2021) extend the GPLVM to handle multiple modalities.

Let $\mathbf{X} = \{\mathbf{x}_n\}_{n=1}^N \in \mathbb{R}^{N \times D}$ and $\mathbf{Y} = \{\mathbf{y}_n\}_{n=1}^N \in \mathbb{R}^{N \times H}$ denote two modalities observed for the same $N$ items. The shared latent representations are denoted by $\mathbf{Z} = \{\mathbf{z}_n\}_{n=1}^N \in \mathbb{R}^{N \times Q}$, where $Q \ll D, H$.

Each dimension of the observed data is modeled independently using Gaussian processes:

$$\mathbf{X}_{:,d} \sim \mathcal{N}(\mathbf{0}, \mathbf{K}^{(\mathbf{X})}), \quad \mathbf{Y}_{:,h} \sim \mathcal{N}(\mathbf{0}, \mathbf{K}^{(\mathbf{Y})}), \tag{2}$$

where $\mathbf{K}^{(\mathbf{X})}$ and $\mathbf{K}^{(\mathbf{Y})}$ are kernel matrices computed over the latent variables $\mathbf{Z}$ for each modality.

In MRD, for each modality $\chi \in \{\mathbf{X}, \mathbf{Y}\}$, the Automatic Relevance Determination (ARD) kernel is used:

$$k_{\mathrm{ARD}}(\mathbf{z}_i, \mathbf{z}_j; \mathbf{w}^{(\chi)}) = \exp\left(-\frac{1}{2}\sum_{q=1}^Q w_q^{(\chi)}(\mathbf{z}_{iq} - \mathbf{z}_{jq})^2\right), \quad w_q^{(\chi)} \in \mathbf{w}^{(\chi)}. \tag{3}$$

This allows each modality to focus on the latent dimensions most relevant to its structure. Figure 3(b) depicts the graphical model of MRD.

However, SGPLVM and MRD have several limitations with respect to our goals. First, they do not explicitly represent user preferences as vectors. Second, the latent space is not constrained to two or three dimensions, often necessitating post hoc visualization using methods such as t-SNE. Third, they do not account for the heterogeneous nature of the data, including the difference between real-valued features and binary ratings, as well as the imbalance and sparsity of user ratings.

The first and second limitations are addressed by RFFs (see Section 2.3) and are discussed in Section 3.1. The third issue is discussed in detail in Section 3.2.

## 2.3 Random Fourier Features

Random Fourier Features (RFFs) (Rahimi & Recht, 2007; Liu et al., 2021; Gundersen et al., 2021; Zhang et al., 2023) offer a scalable approximation to kernel functions by avoiding costly matrix inversion, enabling the application of Gaussian processes to large-scale data.

The RFF transformation $\varphi : \mathbb{R}^Q \to \mathbb{R}^M$ is defined as:

$$\varphi(\mathbf{z}) = \sqrt{\frac{2}{M}} \left( \cos(\boldsymbol{\omega}_1^\top \mathbf{z} + b_1), \cos(\boldsymbol{\omega}_2^\top \mathbf{z} + b_2), \ldots, \cos(\boldsymbol{\omega}_M^\top \mathbf{z} + b_M) \right)^\top, \tag{4}$$

where $\boldsymbol{\omega}_m \sim \mathcal{N}(0, \mathbf{I}_Q)$ and $b_m \sim \mathrm{Unif}(0, 2\pi)$. Under this construction, the inner product $\varphi(\mathbf{z}_i)^\top \varphi(\mathbf{z}_j)$ approximates the RBF kernel with unit parameters ($\theta_1 = \theta_2 = 1$):

$$\varphi(\mathbf{z}_i)^\top \varphi(\mathbf{z}_j) \approx \exp\left( -\frac{\|\mathbf{z}_i - \mathbf{z}_j\|^2}{2} \right) \tag{5}$$

$$= k_{\mathrm{RBF}}(\mathbf{z}_i, \mathbf{z}_j; \theta_1 = \theta_2 = 1). \tag{6}$$

RFFs also support efficient approximation of nonlinear functions. Given observations $\{(\mathbf{z}_n, y_n)\}_{n=1}^N$, a nonlinear function $f$ can be approximated as follows (Watson, 1964; Hastie et al., 2009):

$$f(\mathbf{z}) \approx \sum_{n=1}^N y_n \frac{k(\mathbf{z}_n, \mathbf{z})}{\sum_{j=1}^N k(\mathbf{z}_j, \mathbf{z})} \tag{7}$$

$$\approx \underbrace{\sum_{n=1}^N y_n \frac{\varphi(\mathbf{z}_n)^\top}{\sum_{j=1}^N k(\mathbf{z}_j, \mathbf{z})}}_{\mathbf{w}^\top} \varphi(\mathbf{z}) \tag{8}$$

$$= \mathbf{w}^\top \varphi(\mathbf{z}), \tag{9}$$

where $\mathbf{w} \in \mathbb{R}^M$ is a weight vector. The vector $\mathbf{w}$ is optimized via gradient-based methods, as described in Section 3, with each update requiring $\mathcal{O}(NM)$ computation.

The quality of the RFF approximation depends on the number of features $M$. Theoretical error bounds are given by (Liu et al., 2020; Sutherland & Schneider, 2015):

$$\epsilon = \mathcal{O}\left( \frac{1}{\sqrt{M}} \right), \quad \sup_{\mathbf{z}_i, \mathbf{z}_j \in \mathbf{Z}} \left| \varphi(\mathbf{z}_i)^\top \varphi(\mathbf{z}_j) - k(\mathbf{z}_i, \mathbf{z}_j) \right| \le \epsilon. \tag{10}$$

These bounds are often loose, and reasonable accuracy is achievable with smaller $M$.

## 3 Random Fourier Feature Shared Latent Variable Models

We propose the **Random Fourier Feature Shared Latent Variable Model (RFSLVM)**, a probabilistic generative model that extends MRD by incorporating RFFs. A graphical representation of RFSLVM is shown in Figure 3(c), and the notations used in this section are summarized in Table 2.

RFSLVM models two modalities: item features $\mathbf{X} \in \mathbb{R}^{N \times D}$ and user ratings $\mathbf{Y} \in \{0, 1, \mathrm{NA}\}^{N \times H}$. The goal is to infer a shared low-dimensional **visualization space** $\mathbf{Z} \in \mathbb{R}^{N \times Q}$ and a set of **preference vectors** $\mathbf{W}^{(Y)} \in \mathbb{R}^{H \times M}$ that characterize user preferences.

### 3.1 Generative Process

The generative process of RFSLVM consists of the following four steps:

Table 2: **Notations used in RFSLVM**.

| Notation | Description |
|---|---|
| $\mathbf{X} \in \mathbb{R}^{N \times D}$ | Item feature matrix; each row $\mathbf{x}_n \in \mathbb{R}^D$ represents item $n$. |
| $\mathbf{Y} \in \{0, 1, \text{NA}\}^{N \times H}$ | User rating matrix. 1: positive, 0: negative, NA: unrated. |
| $\mathbf{Z} \in \mathbb{R}^{N \times Q}$ | Latent positions of items in the visualization space; $Q = 2$ for visualization. |
| | $\mathbf{z}_n \in \mathbb{R}^Q$ denotes the latent representation of item $n$. |
| $\mathbf{W}^{(\chi)} \in \mathbb{R}^{\dim(\chi) \times M}$ | Weight matrix for modality $\chi \in \{\mathbf{X}, \mathbf{Y}\}$. $\dim(\mathbf{X}) = D, \dim(\mathbf{Y}) = H$. |
| | $\mathbf{W}^{(\mathbf{X})} = \{\mathbf{w}_d^{(\mathbf{X})}\}_{d=1}^D, \mathbf{W}^{(\mathbf{Y})} = \{\mathbf{w}_h^{(\mathbf{Y})}\}_{h=1}^H$. |
| $\varphi : \mathbb{R}^Q \to \mathbb{R}^M$ | RFF transformation; $\varphi(\mathbf{z}) = \varphi(\mathbf{z}; \mathbf{\Omega}, \mathbf{b})$. |
| | $\mathbf{z} \in \mathbb{R}^Q, \mathbf{\Omega} = \{\boldsymbol{\omega}_m \in \mathbb{R}^Q\}_{m=1}^M, \mathbf{b} = \{b_m \in [0, 2\pi]\}_{m=1}^M$. |
| $\lambda_X, \lambda_Y > 0$ | Regularization coefficients for each modality. |
| $\eta_Z, \eta_X, \eta_Y > 0$ | Learning rates for latent positions and weight matrices. |
| $\kappa_{nh} \geq 0$ | Modality balancing factor for user rating entries. |

**Step 1: Visualization Space.** Each item is associated with a latent position $\mathbf{z}_n$ in a visualization space, sampled from a standard normal distribution:

$$\mathbf{z}_n \sim \mathcal{N}(\mathbf{0}, \mathbf{I}_Q), \quad n = 1, \dots, N. \tag{11}$$

The latent dimensionality is fixed at $Q = 2$ to ensure interpretable visualizations.

**Step 2: Random Fourier Features.** Each latent position $\mathbf{z}_n$ is mapped to a high-dimensional feature space via the RFF transformation:

$$\varphi(\mathbf{z}) \equiv \varphi(\mathbf{z}; \mathbf{\Omega}, \mathbf{b}), \quad \mathbf{z} \in \mathbf{Z}, \tag{12}$$

where $\mathbf{\Omega} = \{\boldsymbol{\omega}_1, \dots, \boldsymbol{\omega}_M\}$ and $\mathbf{b} = \{b_1, \dots, b_M\}$ are sampled as:

$$\boldsymbol{\omega}_m \sim \mathcal{N}(\mathbf{0}, \mathbf{I}_Q), \quad b_m \sim \text{Uniform}(0, 2\pi), \quad m = 1, \dots, M. \tag{13}$$

where $\text{Uniform}(0, 2\pi)$ denotes the uniform distribution over $[0, 2\pi]$. Sampled $\mathbf{\Omega}$ and $\mathbf{b}$ are fixed during training and not optimized.

**Step 3: User Ratings.** Each user $h$ is associated with a **preference vector** $\mathbf{w}_h^{(\mathbf{Y})}$ in the high-dimensional space:

$$\mathbf{w}_h^{(\mathbf{Y})} \sim \mathcal{N}(\mathbf{0}, \lambda_Y^{-1}\mathbf{I}_M). \tag{14}$$

The binary rating $Y_{n,h}$ for item $n$ by user $h$ is drawn from a Bernoulli distribution:

$$Y_{n,h} \sim \text{Bernoulli}\left(\sigma\left(\mathbf{w}_h^{(\mathbf{Y})\top}\varphi(\mathbf{z}_n)\right)\right), \quad n = 1, \dots, N, \quad h = 1, \dots, H, \tag{15}$$

where $\sigma(x) = 1/(1 + e^{-x})$. Here, $\mathbf{w}_h^{(\mathbf{Y})}$ is a linear weight vector in the RFF-induced feature space, which is used to represent a user's nonlinear preference function.

**Step 4: Item Features.** Each feature dimension $d$ of a weight vector $\mathbf{w}_d^{(\mathbf{X})}$:

$$\mathbf{w}_d^{(\mathbf{X})} \sim \mathcal{N}(\mathbf{0}, \lambda_X^{-1}\mathbf{I}_M). \tag{16}$$

The feature value $X_{n,d}$ for item $n$ and dimension $d$ is drawn as:

$$X_{n,d} \sim \mathcal{N}\left(\mathbf{w}_d^{(\mathbf{X})\top}\varphi(\mathbf{z}_n), 1\right), \quad n = 1, \dots, N, \quad d = 1, \dots, D. \tag{17}$$

Here, $\mathbf{w}_d^{(\mathbf{X})}$ is a linear weight vector in the RFF-induced feature space.

### 3.2 Objective Function and Optimization

RFSLVM is trained by minimizing the negative log-likelihood of both modalities. The loss function is defined as:

$$\mathcal{L}(\mathbf{Z}, \mathbf{W^{(X)}}, \mathbf{W^{(Y)}}) = \mathcal{L}(\mathbf{Z}, \mathbf{W^{(X)}}) + \mathcal{L}(\mathbf{Z}, \mathbf{W^{(Y)}}). \tag{18}$$

The first term corresponds to the reconstruction error of item features:

$$\mathcal{L}(\mathbf{Z}, \mathbf{W^{(X)}}) = \sum_{n=1}^{N} \sum_{d=1}^{D} \left( X_{n,d} - \mathbf{w}_d^{(X)\top} \varphi(\mathbf{z}_n) \right)^2 + \lambda_X \sum_{d=1}^{D} \|\mathbf{w}_d^{(X)}\|^2. \tag{19}$$

The second term corresponds to a binary cross-entropy loss for user ratings:

$$\mathcal{L}(\mathbf{Z}, \mathbf{W^{(Y)}}) = -\sum_{n=1}^{N} \sum_{h=1}^{H} \kappa_{nh} \left\{ Y_{n,h} \log(f_{nh}) + (1 - Y_{n,h}) \log(1 - f_{nh}) \right\} \tag{20}$$

$$+ \lambda_Y \sum_{h=1}^{H} \|\mathbf{w}_h^{(Y)}\|^2, \tag{21}$$

where $f_{nh} = \sigma \left( \mathbf{w}_h^{(Y)\top} \varphi(\mathbf{z}_n) \right)$ denotes the predicted probability that user $h$ positively rates item $n$.

The weighting factor $\kappa_{nh}$ adjusts the contribution of each rating to mitigate label imbalance and normalize the loss across modalities:

$$\kappa_{nh} = \frac{ND}{\left( \sum_{n=1}^{N} \sum_{h=1}^{H} \kappa'_{nh} \right) H} \cdot \kappa'_{nh}, \quad \kappa'_{nh} = \begin{cases} 1 & \text{if } Y_{n,h} = 0, \\ \frac{|\{Y_{n,h}=0\}|}{|\{Y_{n,h}=1\}|} & \text{if } Y_{n,h} = 1, \\ 0 & \text{otherwise.} \end{cases} \tag{22}$$

This formulation balances the influence of each label class on the loss function despite severe label imbalance, and normalizes for modality size, thereby promoting stable optimization under sparsity and heterogeneity. The full derivation of the objective function is provided in Appendix A.

To optimize RFSLVM, we minimize the objective in Equation 18 using gradient-based methods. Specifically, we adopt the ADAM optimizer (Kingma & Ba, 2017) in Optax (DeepMind et al., 2020), with auto-differentiation provided by JAX (Bradbury et al., 2018).

## 4 Evaluation

We evaluate RFSLVM on three tasks using publicly available datasets: prediction of binary user ratings in Section 4.2, reconstruction of item features in Section 4.3, and assessment of computational efficiency during inference in Section 4.4. Our evaluation is designed to assess whether RFSLVM achieves performance comparable to representative methods. Such comparability indicates that the learned latent variables can be used for downstream tasks in Section 5.

### 4.1 Datasets

We selected three publicly available datasets with item-level textual descriptions, namely MIND, ANIME, and BEER, as they provide rich lexical information that enables more detailed item feature construction and supports fine-grained interpretation of user preferences. Each dataset was preprocessed to produce binarized user ratings and item features derived from textual metadata, as described below.

**MIND** (Wu et al., 2020) is a large-scale dataset containing user interactions with news articles. User interactions were binarized by treating *clicked* articles as positive ratings and articles that were *displayed but*

Table 3: **Statistics of Evaluation Datasets**. # Ratings is the number of user ratings. # Pos is the number of positive ratings. # Neg is the number of negative ratings. # Items is the number of items $N$. # Users is the number of users $H$.

| Dataset | # Ratings | # Pos | # Neg | # Items | # Users |
|---------|-----------|-------|-------|---------|---------|
| MIND | 768,414 | 43,606 | 724,808 | 3,523 | 2,498 |
| ANIME | 957,949 | 289,030 | 668,919 | 6,269 | 2,978 |
| BEER | 881,724 | 239,001 | 642,723 | 9,244 | 1,859 |

*not clicked* as negative ratings, while others were treated as unrated. We retained users with at least 10 ratings in each class (positive and negative) to ensure sufficient signal per user. Item features were represented using 384-dimensional text embeddings computed from the concatenation of the *title* and *abstract* fields, using the `all-MiniLM-L6-v2` model from the Sentence-Transformers library (Reimers & Gurevych, 2019).

**ANIME**[1] is a dataset consisting of user ratings for various anime. Ratings were binarized by treating scores greater than or equal to 9 as positive ratings, scores less than 9 as negative ratings, and those not scored as unrated. We retained users with at least 10 ratings in each class (positive and negative) to ensure sufficient signal per user. Item features were represented using 384-dimensional text embeddings generated from the *synopsis* field using Sentence-Transformers.

**BEER** (McAuley et al., 2012; McAuley & Leskovec, 2013) is a dataset consisting of user ratings for various beers. Ratings were binarized by treating scores greater than or equal to 4.5 as positive, scores below 4.5 as negative, and unscored items as unrated. We retained users with at least 10 ratings in each class (positive and negative) to ensure sufficient signal per user. Item features were represented using 384-dimensional text embeddings generated by concatenating 30 sampled reviews from the *review/text* field and embedding the result using Sentence-Transformers.

Table 3 summarizes the key statistics of the datasets.

## 4.2 Prediction of User Ratings

We evaluate the predictive performance of RFSLVM on binary classification tasks, comparing it against Logistic Regression (LR), $k$-Nearest Neighbors (kNN), Gaussian Process Classifiers (GPC), MRD, Multi-Modal VAE (MMVAE) (Shi et al., 2019), Neural Collaborative Filtering (NCF) (Rendle et al., 2020), and xDeepFM (Lian et al., 2018). For the multimodal models MRD, MMVAE, and RFSLVM, the dimensionality of the shared latent space was fixed at 2.

We used implementations from (Pedregosa et al., 2011) for LR, kNN, and GPC; from (Argyriou et al., 2020) for NCF and xDeepFM; (GPy, since 2012) for MRD; and from (Senellart et al., 2023) for MMVAE.

We used two standard metrics for imbalanced binary classification: ROC-AUC, which measures the ability to rank positive instances above negative ones (higher is better), and log loss, which quantifies the quality of probabilistic predictions (lower is better).

We used an 80:10:10 split for training, validation, and testing, selecting hyperparameters based on validation performance. The full hyperparameter settings are listed in Appendix B.

Table 4 reports the best-performing results for each model. RFSLVM shows competitive performance compared to the baseline models. Compared to MRD, RFSLVM consistently achieves lower log loss while maintaining comparable ROC-AUC.

---

[1] `https://www.kaggle.com/datasets/dbdmobile/myanimelist-dataset`

[1] GPC yields near-random ROC-AUC ($\sim$0.5) on all datasets, which may reflect difficulty in handling imbalanced and sparse binary ratings.

[2] We evaluated MRD by applying a sigmoid to the predictive mean from the original Gaussian likelihood for $\mathbf{Y}$, and computed ROC-AUC and log loss.

Table 4: **User Rating Prediction Results**. Higher ROC-AUC and lower log loss indicate better predictive performance. Results are reported as mean ± standard deviation over 5 runs with different random seeds. Rounded to $1 \times 10^{-3}$ precision. For each metric, the best score is highlighted in bold within each group, based on whether the model is multimodal or not. MM: Supports multiple modalities. PV: The model provides preference vectors. ✓: supported; ✗: not explicitly supported.

| Model | MM | PV | MIND ROC-AUC | MIND Log loss | ANIME ROC-AUC | ANIME Log loss | BEER ROC-AUC | BEER Log loss |
|---|---|---|---|---|---|---|---|---|
| LR | ✗ | ✓ | $0.639 \pm 0.002$ | $0.36 \pm 0.004$ | $0.678 \pm 0.001$ | $1.264 \pm 0.005$ | $0.591 \pm 0.001$ | $1.331 \pm 0.009$ |
| kNN | ✗ | ✗ | $0.637 \pm 0.003$ | $0.904 \pm 0.016$ | $0.717 \pm 0.001$ | $0.887 \pm 0.004$ | $0.622 \pm 0.001$ | $0.958 \pm 0.013$ |
| GPC [1] | ✗ | ✗ | $0.501 \pm 0.002$ | $0.693 \pm 0.000$ | $0.548 \pm 0.019$ | $0.657 \pm 0.007$ | $0.500 \pm 0.000$ | $0.693 \pm 0.000$ |
| NCF | ✗ | ✓ | $\mathbf{0.685} \pm 0.002$ | $0.245 \pm 0.004$ | $\mathbf{0.795} \pm 0.004$ | $\mathbf{0.585} \pm 0.014$ | $\mathbf{0.724} \pm 0.002$ | $0.642 \pm 0.003$ |
| xDeepFM | ✗ | ✓ | $0.632 \pm 0.004$ | $\mathbf{0.218} \pm 0.002$ | $0.736 \pm 0.002$ | $0.609 \pm 0.001$ | $0.676 \pm 0.001$ | $\mathbf{0.582} \pm 0.002$ |
| MMVAE | ✓ | ✗ | $0.470 \pm 0.055$ | $0.694 \pm 0.001$ | $0.464 \pm 0.044$ | $0.693 \pm 0.000$ | $0.464 \pm 0.092$ | $0.695 \pm 0.002$ |
| MRD [2] | ✓ | ✗ | $\mathbf{0.697} \pm 0.003$ | $0.716 \pm 0.000$ | $\mathbf{0.792} \pm 0.003$ | $0.722 \pm 0.001$ | $0.654 \pm 0.002$ | $0.754 \pm 0.001$ |
| **RFSLVM** | ✓ | ✓ | $0.683 \pm 0.002$ | $\mathbf{0.573} \pm 0.002$ | $0.751 \pm 0.003$ | $\mathbf{0.581} \pm 0.003$ | $\mathbf{0.709} \pm 0.003$ | $\mathbf{0.614} \pm 0.001$ |

Table 5: **Reconstruction Errors**. MAE and RMSE are reported for each model. Lower values indicate better reconstruction performance. Results are reported as mean ± standard deviation over 5 runs with different random seeds. Rounded to $1 \times 10^{-3}$ precision. For each metric, the best score is highlighted in bold.

| Model | MIND MAE | MIND RMSE | ANIME MAE | ANIME RMSE | BEER MAE | BEER RMSE |
|---|---|---|---|---|---|---|
| PCA | $0.782 \pm 0.000$ | $0.980 \pm 0.000$ | $0.771 \pm 0.000$ | $0.968 \pm 0.000$ | $0.773 \pm 0.000$ | $0.973 \pm 0.000$ |
| MMVAE | $0.782 \pm 0.001$ | $0.980 \pm 0.001$ | $0.779 \pm 0.001$ | $0.979 \pm 0.002$ | $0.777 \pm 0.002$ | $0.978 \pm 0.002$ |
| MRD | $0.741 \pm 0.001$ | $0.931 \pm 0.001$ | $0.743 \pm 0.001$ | $0.937 \pm 0.001$ | $0.740 \pm 0.001$ | $0.934 \pm 0.001$ |
| **RFSLVM** | $\mathbf{0.739} \pm 0.001$ | $\mathbf{0.927} \pm 0.001$ | $\mathbf{0.734} \pm 0.001$ | $\mathbf{0.923} \pm 0.001$ | $\mathbf{0.734} \pm 0.001$ | $\mathbf{0.926} \pm 0.001$ |

### 4.3 Reconstruction of Item Features

We evaluated how well the learned representations preserved item information by comparing the observed item features $\mathbf{X}$ with the reconstructed features $\mathbf{X}'$. We compared MRD, MMVAE, and RFSLVM as multimodal models and PCA as a simple baseline.

We reported both Mean Absolute Error (MAE) and Root Mean Squared Error (RMSE), where lower values indicated better reconstruction performance. Table 5 shows the reconstruction errors for item features. RFSLVM achieved competitive reconstruction accuracy across all datasets, attaining the lowest MAE and RMSE on ANIME and BEER.

Figure 4 visualizes the latent spaces inferred by each model. Colors indicate item categories assigned in the MIND dataset. All models capture meaningful structure in the latent space. RFSLVM and MRD exhibit similar patterns in their latent representations.

### 4.4 Processing Time of Inference

We compared the inference time of RFSLVM against MRD, an exact GPLVM that uses full GP inference, and a sparse GPLVM that employs inducing points for scalability (Quiñonero-Candela & Rasmussen, 2005).

To assess scalability, we varied the dataset size $N \in \{100, 300, 500, 1000, 3000, 5000\}$. For each setting, we generated synthetic datasets $\mathbf{X} \in \mathbb{R}^{N \times 384}$ and $\mathbf{Y} \in \{0, 1\}^{N \times 1}$ with randomly sampled values. Inference time was measured per 100 optimization iterations. For MRD and the sparse GPLVM, we used 32 inducing points. For RFSLVM, the number of RFF was fixed at $M = 128$.

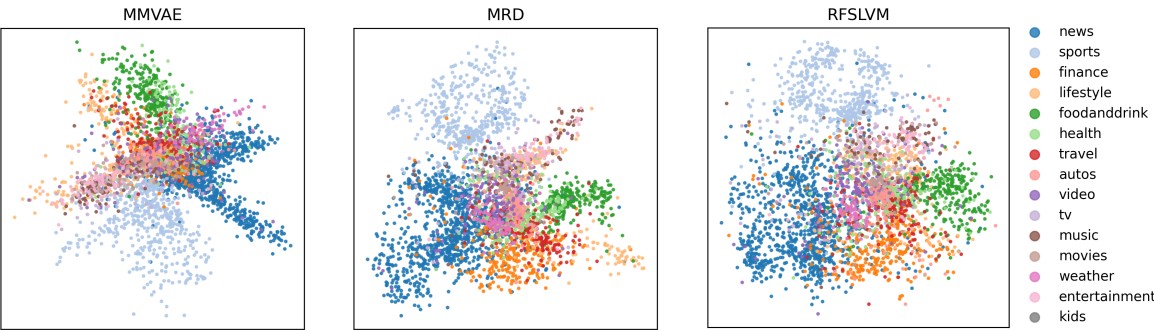

Figure 4: **Visualization Spaces of Each Model**. Each point represents an item in the two-dimensional latent space. Colors indicate item categories assigned in the MIND dataset. This visualization helps us grasp the overall structure of the visualization space.

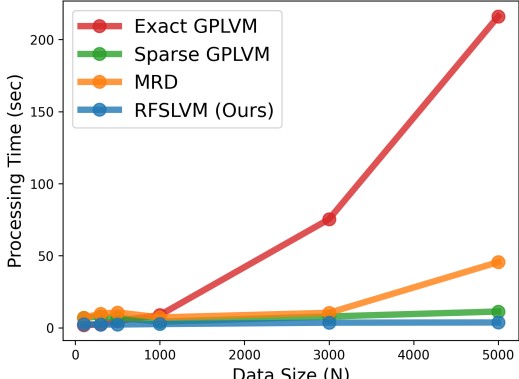

Figure 5: **Processing Time of Inference**. Inference time per 100 optimization iterations as a function of dataset size $N$.

All experiments were conducted on a machine running Ubuntu 22.04.2 LTS, equipped with an AMD Ryzen 7 5800X CPU, an NVIDIA GeForce RTX 3090 GPU, 64 GB RAM, and 24 GB VRAM.

Figure 5 shows the inference time. The exact GPLVM shows cubic time complexity $\mathcal{O}(N^3)$, resulting in rapidly increasing computation time. In contrast, RFSLVM scales efficiently with dataset size, performing better than the sparse GPLVM and MRD in terms of computational cost.

## 4.5  Discussion

**Predictive performance under imbalance:** GPC often struggle with imbalanced data, resulting in ROC-AUC values close to random in Table 4. MRD, while achieving relatively high ROC-AUC, suffers from large log loss values due to its Gaussian likelihood, which is not well-suited for binary ratings. RFSLVM, in contrast, attains relatively stable ROC-AUC together with consistently lower log loss, reflecting the benefits of using a logistic likelihood and the weighting scheme $\kappa_{nh}$ to address both modality-size imbalance and label imbalance during optimization.

**Reconstruction performance:** RFSLVM achieves competitive reconstruction errors (MAE and RMSE), attaining the lowest values across all datasets with only modest margins over MRD. These results suggest that the use of random Fourier features enables expressive yet efficient modeling of nonlinear structures, while preserving information across modalities.

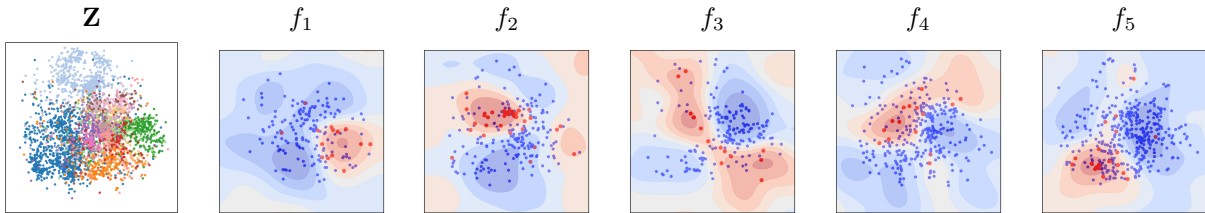

Figure 6: **Visualization Space and User Preference Views**. The visualization space **Z** and five user-specific preference functions $f_1, \ldots, f_5$ are shown. The visualization space is reused from the RFSLVM example in Figure 4. In the preference function views, contour plots represent the values of $f_h(\mathbf{z}) = \sigma(\mathbf{w}_h^{(Y)\top}\varphi(\mathbf{z}))$, with colors transitioning from blue (low preference) to red (high preference). Scatter points indicate items rated by each user (red: positive, blue: negative). By comparing **Z** and $f_h$, we can interpret which regions in the latent space correspond to high or low preferences (see also Figure 2).

**Balanced performance and efficiency:** Compared to exact and sparse GPLVM-based approaches as well as MRD, RFSLVM shows more favorable scaling with dataset size while maintaining competitive predictive and reconstruction accuracy. These results suggest that RFSLVM can be applied to moderate-sized datasets, such as those in Table 3, which contain several thousand items and users along with hundreds of thousands of ratings.

## 5 Analysis Examples

This section presents practical analyses enabled by the visualization space and the preference vectors inferred by RFSLVM. We begin with an overview of the visualization space and user-specific preference views in Section 5.1. Section 5.2 examines user similarity based on preference vectors. Section 5.3 introduces a clustering-based analysis, highlighting representative preferences and identifying promising items within each cluster. All figures and tables in this section are constructed based on learned representations from the MIND dataset.

### 5.1 Visualization Space and User Preference Views

We first illustrate how RFSLVM enables intuitive visualization of user preferences through the learned visualization space. Figure 6 shows the latent item representations and the user-specific preference views. In the figure, the leftmost panel shows the visualization space **Z**. The remaining panels ($f_1$ through $f_5$) depict contour plots of the predicted preference functions for five users. Each user $h$ is associated with a nonlinear preference function defined as

$$f_h(\mathbf{z}) = \sigma(\mathbf{w}_h^{(Y)\top}\varphi(\mathbf{z})), \tag{23}$$

where $\mathbf{w}_h^{(Y)}$ is the user's preference vector and $\varphi(\mathbf{z})$ denotes the RFF transformation. Visualizing user-specific preference functions in the visualization space enables intuitive interpretation of individual user characteristics.

### 5.2 Similarities between User Preferences

RFSLVM enables efficient comparison of complex, nonlinear user preferences by computing similarities between their inferred preference vectors using simple metrics such as inner product or cosine similarity. In this analysis, we identify similar users by ranking the inner products in descending order:

$$\mathbf{w}_i^\top \mathbf{w}_j, \quad i, j \in \{1, \ldots, H\}. \tag{24}$$

Figure 7 lists the five most similar users for each of three selected base users. Each row presents the preference function of a base user (leftmost) alongside those of the five most similar users. These results demonstrate that similar preference patterns can be effectively captured through simple vector operations.

| Base | 1st | 2nd | 3rd | 4th | 5th |
|------|-----|-----|-----|-----|-----|

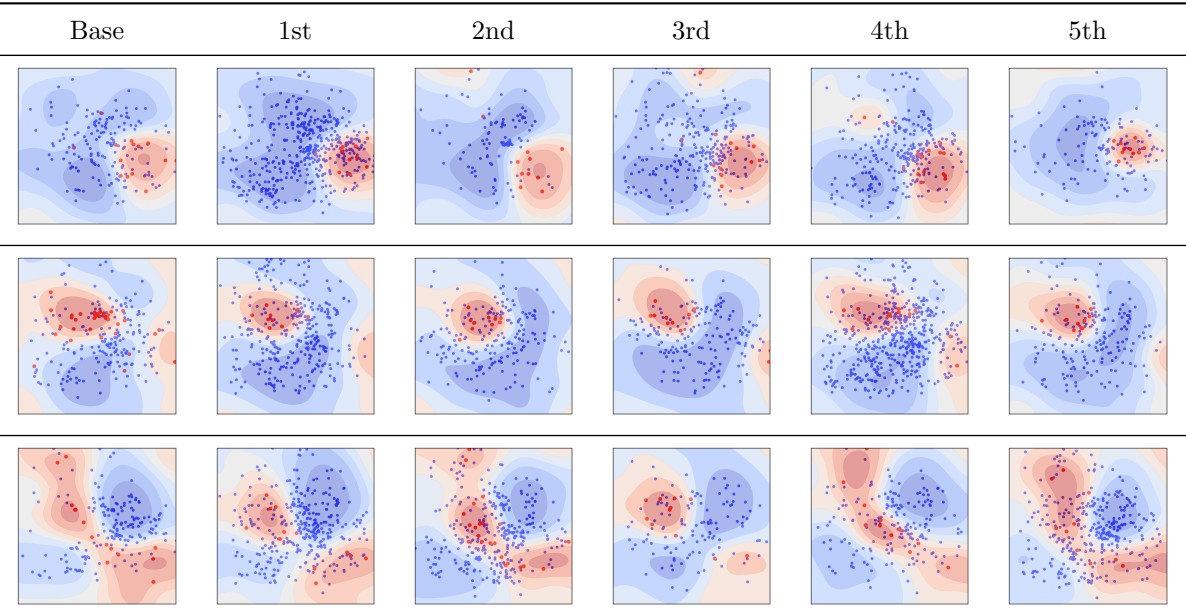

Figure 7: **Similar User Preference Patterns**. For each base user (leftmost column), the five most similar users are identified based on the inner product of preference vectors. Each panel shows the user's preference function $f_h(\mathbf{z}) = \sigma(\mathbf{w}_h^{(Y)\top} \varphi(\mathbf{z}))$ over the visualization space.

### 5.3 Clustering Analysis of User Preferences

Once similarities or distances between user preferences are computed, clustering algorithms can be applied to uncover population-level structures and shared preference patterns. In this analysis, we applied hierarchical clustering to the preference vectors, using cosine similarity. For each cluster $c$, the representative preference function is defined as:

$$f_c(\mathbf{z}) = \sigma\left(\overline{\mathbf{w}}_c^\top \varphi(\mathbf{z})\right), \quad \overline{\mathbf{w}}_c = \frac{1}{|c|} \sum_{h \in c} \mathbf{w}_h^{(Y)}, \tag{25}$$

where $\overline{\mathbf{w}}_c$ denotes the average preference vector of cluster $c$. This formulation enables the computation of cluster-level preference scores. Figure 1 shows the dendrogram introduced in Section 1, constructed from 300 randomly selected users. It illustrates the hierarchical structure of user preferences.

Figure 8 presents five representative clusters, each with its cluster-level preference contours and the top five most promising items, ranked by $f_c(\mathbf{z})$. The table facilitates interpretation of each cluster 's characteristics.

RFSLVM enables cross-modal generation by leveraging a shared latent space across modalities. This allows us not only to analyze observed data, but also to extrapolate into unobserved regions of the visualization space. By evaluating the Equation 25 across the visualization space, we can identify high-preference regions, even for hypothetical items. These regions can then be mapped to item features via the generative process described in Section 3.1, opening new possibilities for content ideation and product design.

## 6 Conclusion

We proposed the **Random Fourier Feature Shared Latent Variable Model (RFSLVM)**, a probabilistic generative model for user preference visualization and analysis. RFSLVM integrates two modalities—real-valued item features and binary user ratings—into a two-dimensional shared latent space (the **visualization space**) and infers user-specific **preference vectors** representing nonlinear preferences. This representation supports both predictive modeling and exploratory analysis.

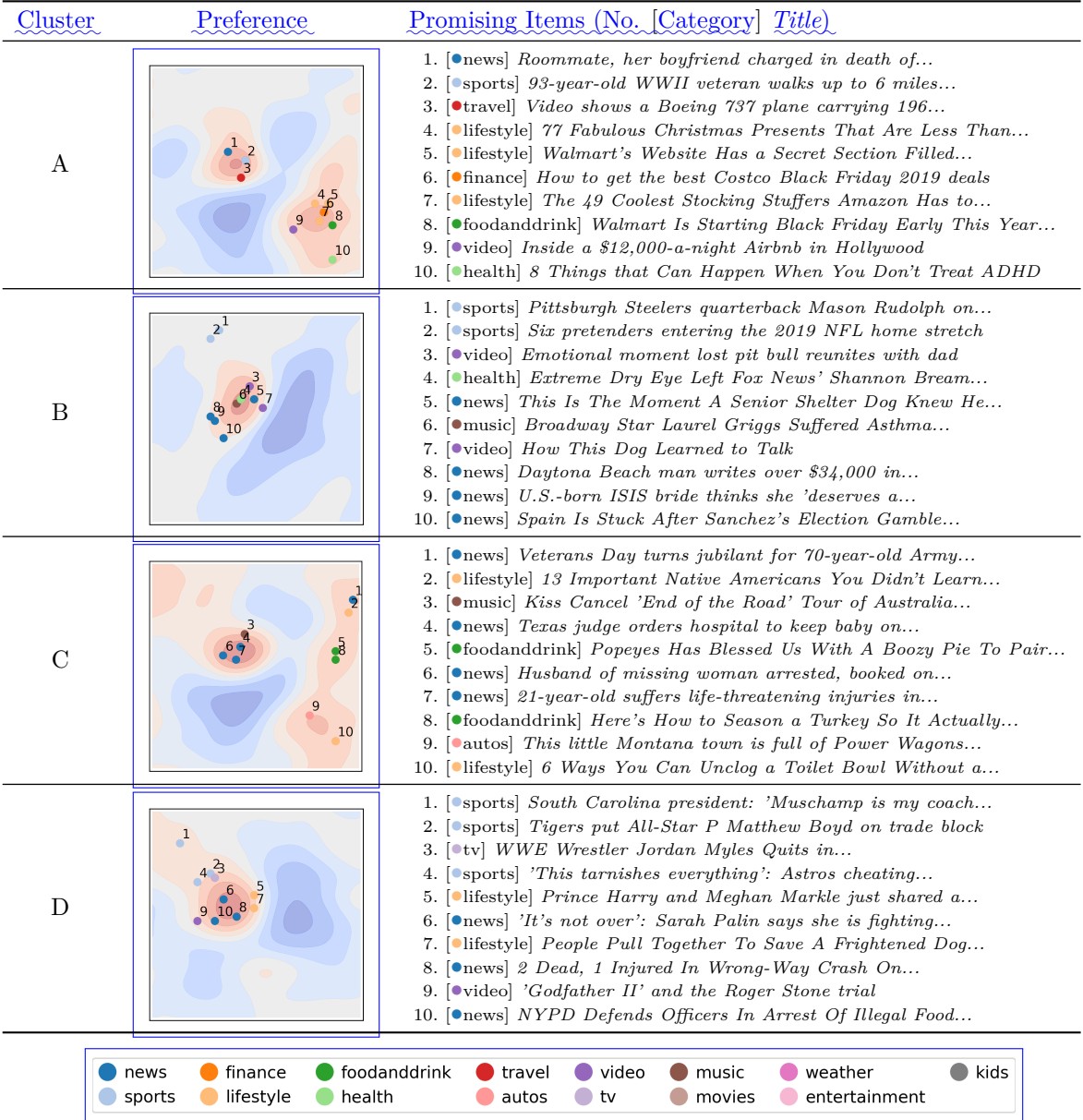

| Cluster | Preference | Promising Items (No. [Category] *Title*) |
|---------|-----------|------------------------------------------|

**Cluster A**
1. [●news] *Roommate, her boyfriend charged in death of...*
2. [●sports] *93-year-old WWII veteran walks up to 6 miles...*
3. [●travel] *Video shows a Boeing 737 plane carrying 196...*
4. [●lifestyle] *77 Fabulous Christmas Presents That Are Less Than...*
5. [●lifestyle] *Walmart's Website Has a Secret Section Filled...*
6. [●finance] *How to get the best Costco Black Friday 2019 deals*
7. [●lifestyle] *The 49 Coolest Stocking Stuffers Amazon Has to...*
8. [●foodanddrink] *Walmart Is Starting Black Friday Early This Year...*
9. [●video] *Inside a $12,000-a-night Airbnb in Hollywood*
10. [●health] *8 Things that Can Happen When You Don't Treat ADHD*

**Cluster B**
1. [●sports] *Pittsburgh Steelers quarterback Mason Rudolph on...*
2. [●sports] *Six pretenders entering the 2019 NFL home stretch*
3. [●video] *Emotional moment lost pit bull reunites with dad*
4. [●health] *Extreme Dry Eye Left Fox News' Shannon Bream...*
5. [●news] *This Is The Moment A Senior Shelter Dog Knew He...*
6. [●music] *Broadway Star Laurel Griggs Suffered Asthma...*
7. [●video] *How This Dog Learned to Talk*
8. [●news] *Daytona Beach man writes over $34,000 in...*
9. [●news] *U.S.-born ISIS bride thinks she 'deserves a...*
10. [●news] *Spain Is Stuck After Sanchez's Election Gamble...*

**Cluster C**
1. [●news] *Veterans Day turns jubilant for 70-year-old Army...*
2. [●lifestyle] *13 Important Native Americans You Didn't Learn...*
3. [●music] *Kiss Cancel 'End of the Road' Tour of Australia...*
4. [●news] *Texas judge orders hospital to keep baby on...*
5. [●foodanddrink] *Popeyes Has Blessed Us With A Boozy Pie To Pair...*
6. [●news] *Husband of missing woman arrested, booked on...*
7. [●news] *21-year-old suffers life-threatening injuries in...*
8. [●foodanddrink] *Here's How to Season a Turkey So It Actually...*
9. [●autos] *This little Montana town is full of Power Wagons...*
10. [●lifestyle] *6 Ways You Can Unclog a Toilet Bowl Without a...*

**Cluster D**
1. [●sports] *South Carolina president: 'Muschamp is my coach...*
2. [●sports] *Tigers put All-Star P Matthew Boyd on trade block*
3. [●tv] *WWE Wrestler Jordan Myles Quits in...*
4. [●sports] *'This tarnishes everything': Astros cheating...*
5. [●lifestyle] *Prince Harry and Meghan Markle just shared a...*
6. [●news] *'It's not over': Sarah Palin says she is fighting...*
7. [●lifestyle] *People Pull Together To Save A Frightened Dog...*
8. [●news] *2 Dead, 1 Injured In Wrong-Way Crash On...*
9. [●video] *'Godfather II' and the Roger Stone trial*
10. [●news] *NYPD Defends Officers In Arrest Of Illegal Food...*

Legend: ● news  ● finance  ● foodanddrink  ● travel  ● video  ● music  ● weather  ● kids
● sports  ● lifestyle  ● health  ● autos  ● tv  ● movies  ● entertainment

Figure 8: **Cluster-level Preferences and Promising Items**. For each cluster (A, B, C, D) shown in Figure 1, the cluster-level preference function $f_c(\mathbf{z})$ is visualized as a contour plot (red = high preference, blue = low). Scatter points indicate the locations of items in the visualization space, randomly sampled from regions with high preferences according to Equation 25, with their category labels and titles listed.

To achieve this, we employed random Fourier features to express preference vectors in a high-dimensional feature space, enabling nonlinear modeling while maintaining interpretability and computational efficiency. The inference algorithm is designed to handle multiple modalities, including differences in data scale, sparsity, and class imbalance.

Empirical evaluations on multiple real-world datasets showed that RFSLVM performs comparably to baseline models in both prediction and reconstruction tasks. We also presented example analyses such as user similarity estimation, clustering, and promising item ranking, which highlight the interpretability and analytical utility of the learned latent representations.

Potential future directions include extensions to broader applications such as recommendation systems (He et al., 2023; Purificato et al., 2024), as well as investigating RFSLVM as a pre-trained model for Bayesian optimization (González-Duque et al., 2024). While this work focuses on binary user ratings, RFSLVM can be extended to handle ordinal preference data by incorporating the Bradley–Terry model (Hino et al., 2010; Caron & Doucet, 2012), which represents another promising avenue for future research. In addition, our experiments confirmed that RFSLVM runs efficiently on datasets of moderate size, involving 1,000–10,000 items or users and 100,000–1,000,000 ratings. Extending the method to substantially larger datasets or to real-time recommendation settings remains an area for further research.

RFSLVM is applicable to domains where both item features and user ratings are available, such as movies, books, restaurants, real estate, and automobiles. We believe this framework enables more interpretable preference modeling and facilitates decision-making through visual and analytical insights.

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

## A  The Objective Function of RFSLVM

We derive the loss function for RFSLVM from the negative log-likelihood of the probabilistic generative process described in Section 3.2 (see Table 2 for notation).

The joint distribution over all variables is factorized as

$$-\log p(\mathbf{X}, \mathbf{Y}, \mathbf{Z}, \mathbf{W^{(X)}}, \mathbf{W^{(Y)}}) \propto -\log \left[ p(\mathbf{X} \mid \mathbf{Z}, \mathbf{W^{(X)}}) \cdot p(\mathbf{W^{(X)}}) \cdot p(\mathbf{Y} \mid \mathbf{Z}, \mathbf{W^{(Y)}}) \cdot p(\mathbf{W^{(Y)}}) \right].$$

We now derive the loss components associated with each modality.

We begin with the item feature modality. Assuming a Gaussian likelihood with unit variance, the negative log-likelihood becomes

$$-\log p(\mathbf{X} \mid \mathbf{Z}, \mathbf{W^{(X)}}) = -\log \prod_{n=1}^{N} \prod_{d=1}^{D} \mathcal{N}(X_{n,d} \mid \mathbf{w}_d^{(\mathbf{X})\top} \varphi(\mathbf{z}_n), 1)$$

$$\propto \sum_{n=1}^{N} \sum_{d=1}^{D} \left( X_{n,d} - \mathbf{w}_d^{(\mathbf{X})\top} \varphi(\mathbf{z}_n) \right)^2.$$

We place a Gaussian prior with zero mean and precision $\lambda_X$ on the item feature weights:

$$-\log p(\mathbf{W}^{(\mathbf{X})}) = -\log \prod_{d=1}^{D} \mathcal{N}(\mathbf{w}_d^{(\mathbf{X})} \mid \mathbf{0}, \lambda_X^{-1}\mathbf{I}_M)$$

$$\propto \lambda_X \sum_{d=1}^{D} \|\mathbf{w}_d^{(\mathbf{X})}\|^2.$$

Combining the above, the loss for the item feature modality is

$$\mathcal{L}(\mathbf{Z}, \mathbf{W}^{(\mathbf{X})}) \propto \sum_{n=1}^{N} \sum_{d=1}^{D} \left( X_{n,d} - \mathbf{w}_d^{(\mathbf{X})\top} \varphi(\mathbf{z}_n) \right)^2 + \lambda_X \sum_{d=1}^{D} \|\mathbf{w}_d^{(\mathbf{X})}\|^2.$$

We next consider the user rating modality. Assuming a Bernoulli likelihood for binary observations, the negative log-likelihood becomes

$$-\log p(\mathbf{Y} \mid \mathbf{Z}, \mathbf{W}^{(\mathbf{Y})}) = -\log \prod_{n=1}^{N} \prod_{h=1}^{H} \text{Bernoulli}\left( Y_{n,h} \mid \sigma(\mathbf{w}_h^{(\mathbf{Y})\top} \varphi(\mathbf{z}_n)) \right)$$

$$\propto -\sum_{n=1}^{N} \sum_{h=1}^{H} \left[ Y_{n,h} \log f_{nh} + (1 - Y_{n,h}) \log(1 - f_{nh}) \right],$$

where $f_{nh} = \sigma(\mathbf{w}_h^{(\mathbf{Y})\top} \varphi(\mathbf{z}_n))$.

A Gaussian prior with zero mean and precision $\lambda_Y$ is placed on the user preference weights:

$$-\log p(\mathbf{W}^{(\mathbf{Y})}) = -\log \prod_{h=1}^{H} \mathcal{N}(\mathbf{w}_h^{(\mathbf{Y})} \mid \mathbf{0}, \lambda_Y^{-1}\mathbf{I}_M)$$

$$\propto \lambda_Y \sum_{h=1}^{H} \|\mathbf{w}_h^{(\mathbf{Y})}\|^2.$$

To address label imbalance and scale discrepancies, we introduce a non-negative weighting factor $\kappa_{nh}$ for each user–item pair. The resulting loss for the user rating modality is

$$\mathcal{L}(\mathbf{Z}, \mathbf{W}^{(\mathbf{Y})}) \propto -\sum_{n=1}^{N} \sum_{h=1}^{H} \kappa_{nh} \left[ Y_{n,h} \log f_{nh} + (1 - Y_{n,h}) \log(1 - f_{nh}) \right] + \lambda_Y \sum_{h=1}^{H} \|\mathbf{w}_h^{(\mathbf{Y})}\|^2.$$

Finally, the total objective function is the sum of the losses from both modalities:

$$\mathcal{L}(\mathbf{Z}, \mathbf{W}^{(\mathbf{X})}, \mathbf{W}^{(\mathbf{Y})}) = \mathcal{L}(\mathbf{Z}, \mathbf{W}^{(\mathbf{X})}) + \mathcal{L}(\mathbf{Z}, \mathbf{W}^{(\mathbf{Y})}).$$

## B  Hyperparameter Settings

We list below the hyperparameter configurations considered for each model. Only the specified hyperparameters are shown; others are set to their default values. Values in curly brackets {} indicate candidate values selected via validation.

- **kNN**
  - $k$: {5, 10}
- **GPC**

- – max_iter_predict: 1,000
- – kernel: {RBF(0.01), RBF(0.1), RBF(1.0)}

- **NCF**
  - – model_name: NeuMF
  - – layer_size: [16, 8, 4]
  - – n_epochs: 10
  - – batch_size: 256
  - – learning_rate: 0.001

- **xDeepFM**
  - – cross_l2, embed_l2, layer_l2: 0.01
  - – cross_layer_size: [20, 10]
  - – init_value: 0.1
  - – epochs: 10, batch_size: 256
  - – learning_rate: 0.001
  - – user_Linear_part: True
  - – user_CIN_part, user_DNN_part: True

- **MMVAE**
  - – latent_dim: 2
  - – max_epochs: 100
  - – learning_rate: 0.001
  - – use_early_stopping: True

- **MRD**
  - – max_iters: 100
  - – latent_dim: 2
  - – num_inducing: {16, 32, 64}

- **RFSLVM**
  - – Iterations $T$: {100, ..., 1000}
  - – Latent dimension $Q$: 2
  - – Random features $M$: {128, 256, 384}
  - – Learning rates: $\eta_Z = 0.01$, $\eta_X = \eta_Y = 0.001$
  - – Regularization $\lambda_X$, $\lambda_Y$: 0.01

## C   Effect of Latent Dimension and Random Feature Size

We investigate how the visualization space dimensionality $Q$ and the number of random Fourier features $M$ affect the predictive performance of RFSLVM on the MIND dataset. Figure 9 shows the ROC-AUC scores for various combinations of $Q \in \{2, 3, 4, 8\}$ and $M \in \{128, 256, 384\}$. Overall, performance remains stable for $Q = 2$ to 4, with a slight degradation observed at $Q = 8$. This suggests that low-dimensional latent spaces are sufficient to capture meaningful user-item relationships.

## D   Effect of Random Fourier Features and Random Seed

Table 6 reports the average ROC-AUC and standard deviations over five different random seeds for varying numbers of random Fourier features $M$. Across datasets and feature sizes, the variance across runs remains very small (typically below 0.005), indicating that RFSLVM is robust to the choice of random seed.

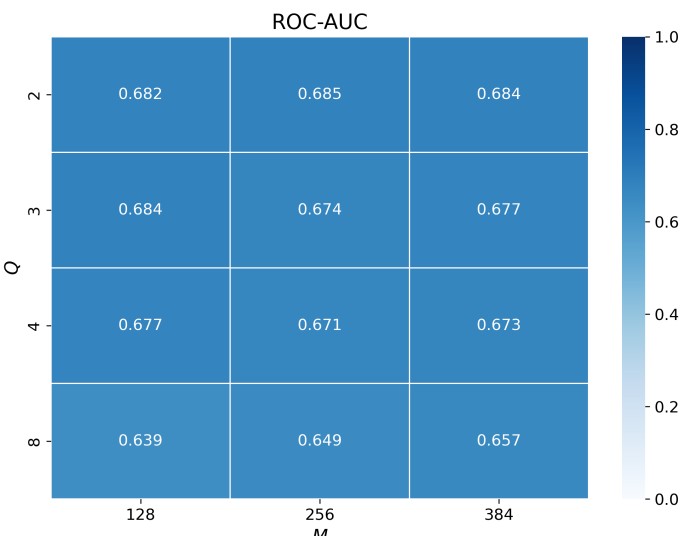

Figure 9: ROC-AUC on the MIND dataset for different combinations of latent dimension $Q \in \{2, 3, 4, 8\}$ and number of random Fourier features $M \in \{128, 256, 384\}$.

Table 6: ROC-AUC (mean ± std. across 5 random seeds) for different numbers of random Fourier features $M$.

| M | MIND | ANIME | BEER |
|---|---|---|---|
| 128 | $0.682 \pm 0.005$ | $0.751 \pm 0.003$ | $0.708 \pm 0.004$ |
| 256 | $0.683 \pm 0.002$ | $0.751 \pm 0.002$ | $0.709 \pm 0.003$ |
| 512 | $0.683 \pm 0.003$ | $0.750 \pm 0.001$ | $0.708 \pm 0.002$ |

## E  Visualization Comparison Across Methods

To complement the quantitative evaluation, we present visualizations on the ANIME and BEER datasets, shown in Figures 10 and 11, obtained by PCA, t-SNE, UMAP, MMVAE, MRD [2], and RFSLVM.

---

[2] Due to the high computational cost of MRD on the full datasets in Table 3, the visualizations shown here are based on a subset of $3,000$ items and $300$ users sampled from the original datasets.

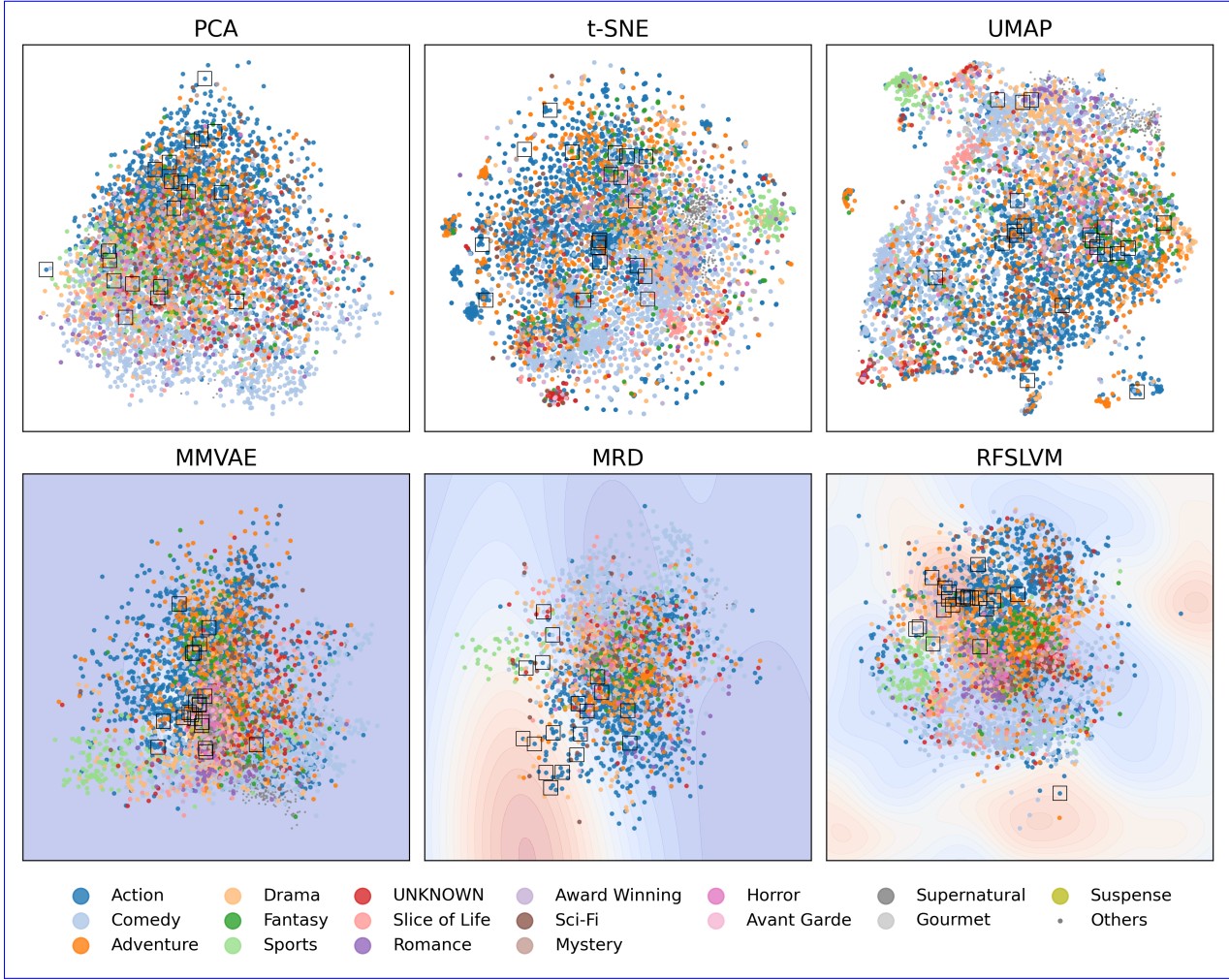

Figure 10: Visualization comparison on the ANIME dataset. Each panel shows the visualization spaces from PCA, t-SNE, UMAP, MMVAE, MRD, and RFSLVM. Points are colored by item category. For the multimodal models (MMVAE, MRD, and RFSLVM), contour plots can be shown to visualize the preference function of a sampled user. Black squares indicate 20 items positively rated by that user. By comparing these items with regions of high preference intensity ( red areas) , preference patterns can be visually examined.

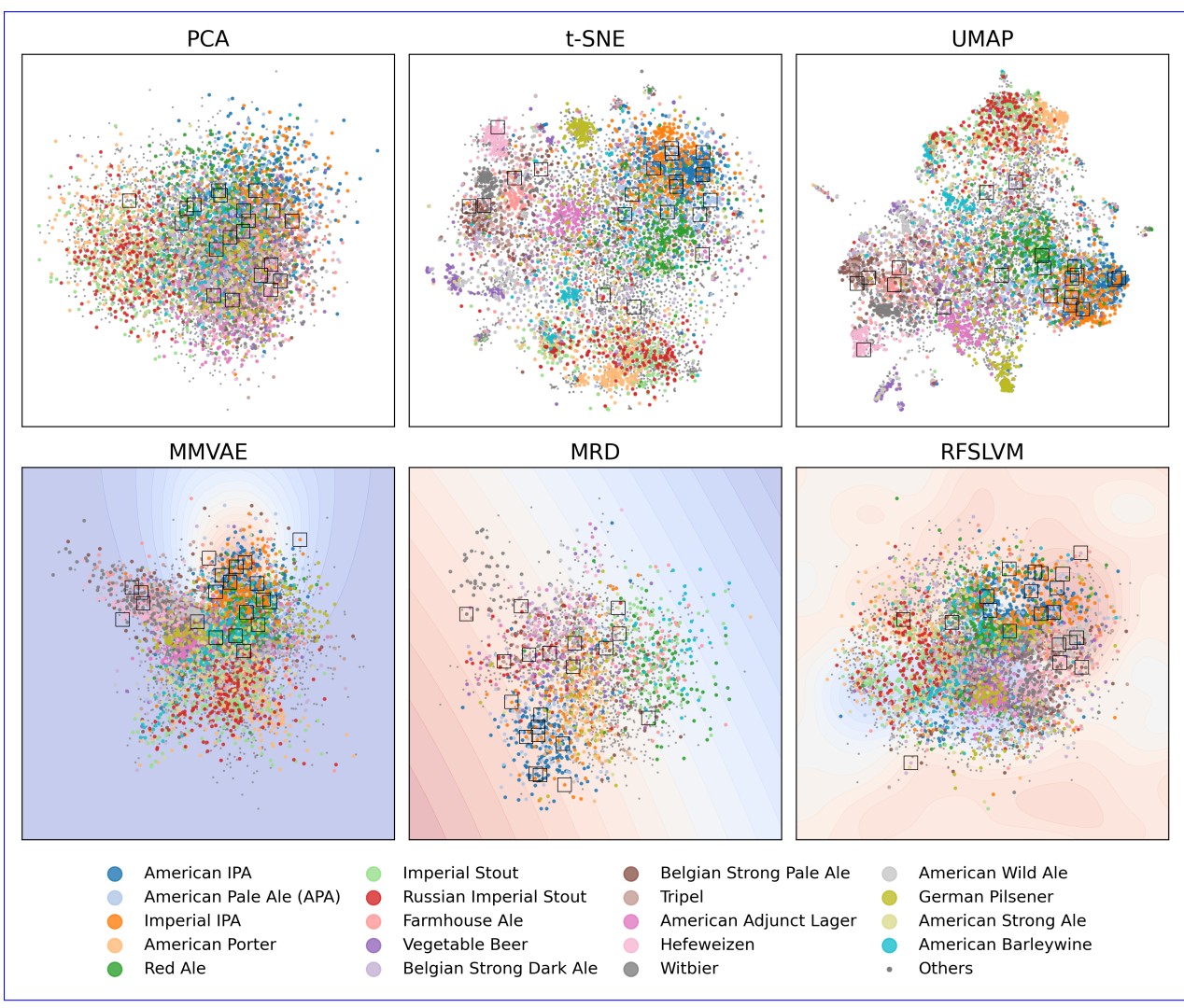

Figure 11: Visualization comparison on the BEER dataset. See Figure 10 for a description of the visualization format.

