# OpenReview forum: "Random Fourier Feature Shared Latent Variable Models for User Preference Visualization and Analysis"
_TMLR — Rejected by TMLR_

### Review · Reviewer_R6br · 2025-08-20

**Summary Of Contributions:**

The paper proposes the **Random Fourier Feature Shared Latent Variable Model (RFSLVM)**, a probabilistic model for user preference modeling that aims to jointly analyze item features and binary user ratings while enabling interpretable 2D visualizations and user preference analysis.

**Key claimed contributions:**

1. **Interpretable 2D Visualization Space**: RFSLVM learns a shared, two-dimensional latent space ("visualization space") for items, allowing direct 2D plotting without post-hoc methods like t-SNE. This is achieved by constraining the latent dimension $ Q = 2 $.

2. **User Preference Vectors**: The model infers user-specific weight vectors $ \mathbf{w}_h^{(Y)} $ in a Random Fourier Feature (RFF)-transformed space, enabling representation of nonlinear preferences. These vectors are intended to support downstream tasks like user similarity computation and clustering.

3. **Scalable Inference via RFFs**: Uses RFFs to approximate Gaussian processes, reducing computational complexity from $ \mathcal{O}(N^3) $ to $ \mathcal{O}(NM) $ per training iteration, where $ M $ is the number of random features.

4. **Heterogeneous Data Handling**: Introduces a modality-balancing factor $ \kappa_{nh} $ to handle sparsity and class imbalance in binary ratings.

The model is evaluated on three datasets (MIND, ANIME, BEER), showing competitive ROC-AUC and reconstruction accuracy against baselines like MRD, MMVAE, NCF, and xDeepFM. The authors demonstrate qualitative analyses such as hierarchical clustering of users (Figure 1) and visualization of cluster-level preference patterns (Table 7).

**Audience:**

Yes

**Broader Impact Concerns:**

/

**Claims And Evidence:**

No

**Requested Changes:**

- Adding more insights and details to Section 4 is needed
- Some of the latest GPLVM literature for multi-view/multi-modal data is missing. (I suspect that some SOTA deep-learning-based methods are also missing)
- Report mean and standard deviation over multiple runs (e.g., 3–5 random seeds) for key metrics in Table 4 and Table 5.
- Consider renaming “preference vectors” to “RFF-space preference weights” to avoid implying semantic interpretability.
-  Consider adding a short discussion in Section 6 about the practical scalability of RFSLVM. Acknowledge that it is suitable for moderate-sized datasets and expert analysis, but not for large-scale, real-time recommendation systems.
- The core model is a combination of existing techniques: MRD for multi-modality, RFFs for scalability, and a fixed 2D latent space for visualization. The authors should more honestly position RFSLVM as an engineering integration rather than a methodological breakthrough.
Please revise the abstract and introduction to tone down claims of significant novelty. Instead, emphasize the practical value of combining these components into a single interpretable framework suitable for expert-in-the-loop applications.

**Strengths And Weaknesses:**

### **Strengths**

- The paper is well-motivated by real-world applications where interpretability is crucial.
- The model explicitly learns a **two-dimensional latent space** ($ Q = 2 $) for items, which enables direct, post-hoc-free visualization. This is a practical advantage over models like MRD or MMVAE that require t-SNE/UMAP for 2D plotting.
- By using RFFs to approximate the RBF kernel, the computational complexity is reduced and the **preference vectors** $ \mathbf{w}_h^{(Y)} \in \mathbb{R}^M $ in RFF space allows for downstream analyses.
- Section 5 is a strong part of the paper. The hierarchical clustering (Figure 1), user similarity analysis (Table 6), and cluster-level item recommendations (Table 7) effectively demonstrate the model’s interpretability and utility for decision-making.

### **Weaknesses**

 - **Incremental Technical Contribution**: The core model is a relatively straightforward combination of existing components:
   - Shared latent space from **MRD**
   - RFF-based approximation from **RFLVM**
   - 2D constraint for visualization

    The use of RFFs to represent nonlinear preferences as linear weights in feature space is well-known (cf. Rahimi & Recht, 2007), and applying it within a shared latent variable framework does not represent a significant conceptual leap.

- The paper presents the preference vectors (PV) $ \mathbf{w}_h^{(Y)} $ as a key contribution, but they are standard in kernel approximation and sparse GP literature. Also, in MRD, can we take $\mathbf{w}^{\cal x}$ as PV similar to RFSLVM?

- **Reconstruction Performance is Not Superior**: RFSLVM achieves slightly better MAE/RMSE than MRD and MMVAE (Table 5), but the differences are marginal. Given the simplicity of a linear autoencoder or PCA, it's unclear if the added complexity is justified.

- The model uses fixed random Fourier features (not learned), but the sensitivity to the number of features $ M $ and random seed is not thoroughly explored. While Appendix C varies $ M $, there is no discussion of variance across runs.

- More insights in Section 4 should be given. The current version reads more like a concise experimental report. For example, it is not clear how GPC is used. Did you train the GP hyperparameters or just fix the kernel as RFSLVM? I am a bit shocked that GPC works so badly.

---

> ### Author Response · Authors · 2026-01-29
>
> We thank the reviewer for the detailed and careful review of our manuscript and for the thorough technical feedback.
> We appreciate the in-depth discussion of both the strengths and limitations of the proposed model, as well as the specific and constructive suggestions for improving the evaluation and positioning of the paper.
>
> Below, we respond to each comment in detail.
> The revised manuscript has already been uploaded.
> In the revised manuscript, changes and newly added content are shown in blue.
> Updated figures are highlighted with blue frames.
>
> ## Requested Changes
>
> > (R-1) Adding more insights and details to Section 4 is needed
>
> We added Section 4.5 (Discussion, pp.11-12), which discusses the following topics:
>
> - Predictive performance under imbalance
> - Reconstruction performance
> - Balanced performance and efficiency
>
> > (R-2) Some of the latest GPLVM literature for multi-view/multi-modal data is missing. (I suspect that some SOTA deep-learning-based methods are also missing)
>
> We added recent and representative works on t-SNE- and UMAP-based visualization methods
> with multimodal extensions, as well as GPLVM-based and deep-learning-based multimodal models,
> to better position RFSLVM within the current literature.
> These additions are discussed in the Introduction (pp.2–3).
>
> Our primary focus is not on achieving state-of-the-art predictive performance,
> but on assessing whether RFSLVM attains performance that is competitive with
> representative and well-established methods.
> This is sufficient for our goal of using the inferred latent representations
> for downstream visualization and analysis.
> This evaluation perspective has been added to the revised Section 4 (Evaluation, p.8).
>
> > (R-3) Report mean and standard deviation over multiple runs (e.g., 3–5 random seeds) for key metrics in Table 4 and Table 5.
>
> We re-ran the experiments over 5 random seeds and updated Tables 4 and 5 (p.10) with mean ± standard deviation.
>
> > (R-4) Consider renaming “preference vectors” to “RFF-space preference weights” to avoid implying semantic interpretability
>
> We clarified at the first mention that the “preference vectors” should be understood as weight vectors in the RFF space.
> (see Abstract, p.1, Introduction, p.3)
>
> > (R-5) Consider adding a short discussion in Section 6 about the practical scalability of RFSLVM. Acknowledge that it is suitable for moderate-sized datasets and expert analysis, but not for large-scale, real-time recommendation systems.
>
> In Section 6 (Conclusion, p.15), we now explicitly acknowledge that RFSLVM is efficient for moderate-sized datasets (1,000–10,000 items or users and 100,000–1,000,000 ratings), while extending it to larger datasets or real-time recommendation remains a topic for future research.
>
> > (R-6) The core model is a combination of existing techniques: MRD for multi-modality, RFFs for scalability, and a fixed 2D latent space for visualization. The authors should more honestly position RFSLVM as an engineering integration rather than a methodological breakthrough. Please revise the abstract and introduction to tone down claims of significant novelty. Instead, emphasize the practical value of combining these components into a single interpretable framework suitable for expert-in-the-loop applications.
>
> We revised the Abstract and Introduction (pp.3-4) to adjust their tone, emphasizing the integration of existing methods for practical use and interpretability.
>
> To further clarify this focus, we also revised the paper title to highlight
> visualization and analysis as the primary focus.

---

> > ### Author Response · Authors · 2026-01-29
> >
> > ## Weaknesses
> >
> > > (W-1) The use of RFFs to represent nonlinear preferences as linear weights in feature space is well-known (cf. Rahimi & Recht, 2007), and applying it within a shared latent variable framework does not represent a significant conceptual leap.
> >
> > This concern relates to the overall positioning of the contribution and is addressed in more detail in our response to (R-6).
> >
> > > (W-2) The paper presents the preference vectors (PV) as a key contribution, but they are standard in kernel approximation and sparse GP literature. Also, in MRD, can we take as PV similar to RFSLVM?
> >
> > In MRD, ARD weights identify relevant latent dimensions at the modality level.
> > As a result, MRD does not provide user-specific vectors that parameterize individual
> > preference functions.
> >
> > > (W-3) Reconstruction Performance is Not Superior: RFSLVM achieves slightly better MAE/RMSE than MRD and MMVAE (Table 5), but the differences are marginal. Given the simplicity of a linear autoencoder or PCA, it's unclear if the added complexity is justified.
> >
> > As shown in Table 5, RFSLVM consistently achieves the lowest reconstruction errors across
> > datasets, with improvements over simple baselines such as PCA and incremental gains
> > relative to MRD.
> > At the same time, RFSLVM maintains predictive performance comparable to baseline methods
> > (Table 4), indicating that the learned latent space supports both reconstruction and
> > preference prediction.
> > This balance supports the interpretation of item–preference relationships in the shared latent space.
> >
> > > (W-4) The model uses fixed random Fourier features (not learned), but the sensitivity to the number of features and random seed is not thoroughly explored. While Appendix C varies, there is no discussion of variance across runs.
> >
> > We conducted additional experiments with five random seeds and observed negligible
> > variance across runs for M ≥ 128 (Appendix D, Table 6).
> >
> > > (W-5) More insights in Section 4 should be given. The current version reads more like a concise experimental report. For example, it is not clear how GPC is used. Did you train the GP hyperparameters or just fix the kernel as RFSLVM? I am a bit shocked that GPC works so badly.
> >
> > As described in our response to (R-1), we added Section 4.5 (Discussion) to provide
> > additional insights into the experimental results, including a discussion of the
> > observed behavior of GPC.

---

> > > ### Comment · Reviewer_R6br · 2026-02-17
> > >
> > > Still not clear to me how GPC was trained and used. Could you explain this part of the paper in more detail?

---

> > > > ### Author Response · Authors · 2026-02-18
> > > >
> > > > We thank the reviewer for prompting this clarification and explain below how GPC was trained and used in our experiments.
> > > >
> > > > **How GPC was trained and used**
> > > >
> > > > For clarity, we summarize the GPC setup (Section 4.2 and Appendix B).
> > > >
> > > > - A separate GPC model was trained independently for each user.
> > > > - Input: 384-dimensional item text embedding (Section 4.1).
> > > > - Target: binary rating (positive vs. negative); unrated entries were excluded.
> > > >
> > > > The original hyperparameter grid (Appendix B) was:
> > > >
> > > > - kernel: {RBF(0.01), RBF(0.1), RBF(1.0)}
> > > > - max_iter_predict: 1,000
> > > >
> > > > ---
> > > >
> > > > **Additional analysis**
> > > >
> > > > We carefully inspected the dataset scale and the optimization behavior.
> > > >
> > > > The mean pairwise distance between item embeddings is ≈27–28.
> > > > With RBF lengthscales 0.01–1.0, kernel values collapse toward zero,
> > > > yielding an almost diagonal kernel matrix and predictions near 0.5.
> > > >
> > > > Thus, the earlier near-random performance was due to a kernel–data scale mismatch.
> > > >
> > > > We re-evaluated GPC with an expanded kernel grid {RBF(10.0), RBF(30.0), RBF(50.0)}
> > > > on a subsample of 100 users (100 rated items per user, preserving the positive/negative ratio), reporting results over five random seeds.
> > > >
> > > > The updated ROC-AUC scores are:
> > > >
> > > > - MIND: 0.623 ± 0.016
> > > > - ANIME: 0.724 ± 0.021
> > > > - BEER: 0.653 ± 0.009
> > > >
> > > > The revised manuscript will reflect this clarification
> > > > (Section 4.2, Section 4.5, Appendix B).

---

### Review · Reviewer_Qys6 · 2025-09-03

**Summary Of Contributions:**

The authors build a generative / latent variable model which handles multimodal features toembed user preferences for the purposes of both visualization and prediction of user ratings.

In a nutshell, their model seeks to reconstruct item features $\mathbf{X}$ and user preference ratings $\mathbf{Y}$ from a low dimensional visualization embedding $\mathbf{Z}$ and two preference vectors $\mathbf{W}^{X}$ and $\mathbf{W}^{Y}$. The use a Gaussian Process latent variable model and incorporate random fourier features as the kernel (helping GPs scale to large data in spite of their computational complexity).

They authors evaluate their model for predicting user preferences primarily across three benchmark datasets (MIND, ANIME, and BEER), highlighting competitive (but not consistently SoTA) performance, as well as subjective visualization spaces and processing/inference times.

**Audience:**

Yes

**Claims And Evidence:**

Yes

**Requested Changes:**

1) could the authors confirm why they did not follow any of the typical recommendation system evaluation paradigms (in terms of datasets, comparison methods, and metrics)?  I admit in advance I'm writing this whilst travelling so if this evaluation was in the supp mat please let me know. If it is, I would nonetheless suggest moving at least some of it to the main text.

2) And similarly, if there is a good reason why this was not done, could the authors comment on whether this is a significant limitation / limits the scope of applicability of their method?

3) Relating to the visualizations (Figure 1 and  6 in particular), if semantic labels can be added to these to relate the visualizations to interpretable concepts in the dataset it would be helpful. It is possible to cluster anything and see a result, the question is whether these clusters are meaningful, and its hard to find the complexity of the dendogram in Figure 1 convincing.

**Strengths And Weaknesses:**

The use of RFF with GPs is a compelling design decision, the benefits of which are clearly illustrated in Figure 4, and the simplicity of the generative model (Figure 2c) also keeps things easy to follow.

The visualizations are also a nice sanity check although *I'm actually not sure what to really take from the comparisons in Figure 3* - they all look reasonable in terms of content separation, in fact the authors' method arguably is not achieving the cleanest separation (but its close).

The paper was easy to follow and well written.

Weaknesses

Besides the point mentioned above, I find the method relatively incremental, and besides the inference time, the performance evaluations also support this interpretation. Similarly, whilst I found Table 1 honest and clear (as much of the paper is), at a top level the incorporation of preference vectors is the 'only' thing (with some caveats relating to constraints on output distribution or latent dimensionality) that separates the authors' method from two of the other comparison methods. None of these things are problems per se, and overall the work is thorough, but these downsides do make the paper slightly underwhelming.

I have to admit, I have some experience with recommendation systems, and maybe the authors can correct me, but it was surprising that the evaluations did not involve any standard recommendation benchmarks or metrics. I understand that the paradigm is slightly different because this system assumes that user preferences exist... But nonetheless, in the same way the authors derive preferences by binarizing features in the benchmark datasets they used, I imagine similar preference scores could be derived for classic recommendation benchmarks like Amazon/Baby etc.

Finally, whilst I appreciated the illustration in Figures 5 and 6 for how the visualization space can, in principle, be used to derive insights relating to user preferences, this all felt relatively abstract. Figure 1, similarly, looked pretty but it is hard to know really what to do with what seems to be a finely grained discretisation of a relatively continuous space.

---

> ### Author Response · Authors · 2026-01-29
>
> We thank the reviewer for the careful reading of our manuscript and for the thoughtful and constructive feedback.
> We appreciate the reviewer’s comments on both the modeling choices and the empirical evaluation, as well as the overall clarity of the paper.
>
> Below, we respond to each concern in detail.
> The revised manuscript has already been uploaded.
> In the revised manuscript, changes and newly added content are shown in blue.
> Updated figures are highlighted with blue frames.
>
> ## Requested Changes
>
> > (R-1) could the authors confirm why they did not follow any of the typical recommendation system evaluation paradigms (in terms of datasets, comparison methods, and metrics)? I admit in advance I'm writing this whilst travelling so if this evaluation was in the supp mat please let me know. If it is, I would nonetheless suggest moving at least some of it to the main text.
>
> We did not use standard recommendation benchmarks such as MovieLens or Amazon,
> as our analysis focuses on interpretable preference visualization based on item-level descriptions,
> which these benchmarks do not natively provide.
> Accordingly, we used MIND, ANIME, and BEER, which include rich textual item descriptions, and clarified this rationale in the revised Section 4.1 (Datasets).
>
> Regarding evaluation metrics, we report ROC-AUC and log loss to examine how the model captures user preference signals, and reconstruction error to evaluate how the learned latent space preserves item information.
> Taken together, these metrics provide insight into how the shared latent space relates items to user preferences.
>
> Ranking-based metrics such as Precision@K are less directly aligned with our objective, which emphasizes preference modeling and analysis rather than top-K recommendation.
>
> > (R-2) And similarly, if there is a good reason why this was not done, could the authors comment on whether this is a significant limitation / limits the scope of applicability of their method?
>
> RFSLVM is not intended to replace large-scale recommendation systems evaluated under standard top-K metrics.
> Rather, it is designed for settings where interpretability and analysis of item–preference relationships are central, such as expert-in-the-loop exploration and preference analysis, which defines the scope of its applicability.
>
> > (R-3) Relating to the visualizations (Figure 1 and 6 in particular), if semantic labels can be added to these to relate the visualizations to interpretable concepts in the dataset it would be helpful. It is possible to cluster anything and see a result, the question is whether these clusters are meaningful, and its hard to find the complexity of the dendogram in Figure 1 convincing.
>
> Figure 1 provides a high-level overview of dominant user preference structures.
> In response to the reviewer’s suggestion, we enhanced Figure 2 (p.3) and Figure 8 (p.14) by visualizing cluster-level preference functions together with representative items annotated with semantic category labels, supporting the interpretation of clusters in terms of dataset semantics.

---

> > ### Author Response · Authors · 2026-01-29
> >
> > ## Weaknesses
> >
> > > (W-1) I find the method relatively incremental, and besides the inference time, the performance evaluations also support this interpretation. Similarly, whilst I found Table 1 honest and clear (as much of the paper is), at a top level the incorporation of preference vectors is the 'only' thing (with some caveats relating to constraints on output distribution or latent dimensionality) that separates the authors' method from two of the other comparison methods. None of these things are problems per se, and overall the work is thorough, but these downsides do make the paper slightly underwhelming.
> >
> > The contribution focuses on providing a unified probabilistic framework in which user-specific preference vectors are explicit and directly usable for visualization and expert-driven analysis.
> >
> > We revised the Abstract, Introduction (p.3), and the paper title to clarify the positioning of RFSLVM as an integration of existing techniques with a focus on visualization and analytical use.
> >
> > > (W-2) I have to admit, I have some experience with recommendation systems, and maybe the authors can correct me, but it was surprising that the evaluations did not involve any standard recommendation benchmarks or metrics. I understand that the paradigm is slightly different because this system assumes that user preferences exist... But nonetheless, in the same way the authors derive preferences by binarizing features in the benchmark datasets they used, I imagine similar preference scores could be derived for classic recommendation benchmarks like Amazon/Baby etc.
> >
> > This point is closely related to (R-1).
> > While it is possible in principle to derive preference labels for standard recommendation benchmarks, our evaluation focuses on datasets that natively provide item-level descriptions to support interpretable preference visualization.
> > This rationale is clarified in Section 4.1 (Datasets, p.8).
> >
> > > (W-3) Finally, whilst I appreciated the illustration in Figures 5 and 6 for how the visualization space can, in principle, be used to derive insights relating to user preferences, this all felt relatively abstract. Figure 1, similarly, looked pretty but it is hard to know really what to do with what seems to be a finely grained discretisation of a relatively continuous space.
> >
> > Figure 1 provides a global overview of user preference structures.
> > Figure 2 relates item categories to preference patterns in the visualization space.
> > Figure 8 examines selected clusters through cluster-level preferences and representative items.
> > Together, these visualizations support exploratory analysis across global, cluster-level,
> > and user-level preferences.
> >
> > The learned preference representations can also serve as an entry point for deeper analyses, for example, examining whether users with similar preference profiles exhibit consistent purchasing or consumption sequences.

---

### Review · Reviewer_Swdp · 2026-01-17

**Summary Of Contributions:**

The authors propose a method called Random Fourier Shared Latent Variable Models.
They apply their method to visualize and analyze users' preferences.
The method use probabilistic machinery to relate data and latent variables.
A claimed originality is multi-modality: the method can related latent variables to two data spaces.
The authors compare their method against others, most of which not having all the same features as the proposed method.

**Audience:**

Yes

**Claims And Evidence:**

Yes

**Requested Changes:**

As said, the paper is well written, well organized, technically correct.
However, it somehow lacks a remarkable contribution.
The authors should perhaps show t-SNE plots of their data sets to be fairer (t-SNE is only reported in quantitative results and tables).
Would something like this be a competitor?
https://lvdmaaten.github.io/multiplemaps/Multiple_maps_t-SNE/Multiple_maps_t-SNE.html
(concatenating X and Y?)
or this: https://arxiv.org/abs/2005.00670 ?
All in all, there is the feeling that the authors have not challenged their method enough, with other competing methods, although it is true that the framework of multimodality might restrict the list of eligible candidates.
Several data sets are assessed quantitatively, whereas only one is illustrated, in a context where visualization is claimed to matter.

**Strengths And Weaknesses:**

The paper is very well written and easy to read.
It is well organized too.
The method is well described, with the key elements in the main text and additional equations and losses in the appendices.
The paper is well illustrated with great care.
The experimental plan is good.
Code is provided as supplementary material and ensures good reproducibility.
On the downsides, the originality of the paper seems to hold mainly in the multimodality and it appears like relatively straightforward extensions of existing methods.
Another weak point is that the probabilistic approach and the addition of multimodality seem to have a cost in terms of visualization power. The strong bias towards cluster gap magnification that is typical of more ad hoc methods like t-SNE and UMAP seems to absent here, with cluttered visualizations showing overlapping clusters.

---

> ### Author Response · Authors · 2026-01-29
>
> We sincerely thank the reviewer for the careful reading of our manuscript and for the constructive comments.
> We appreciate the positive assessment regarding the clarity, organization, and experimental design of the paper.
>
> Below, we respond to each concern in detail.
> Throughout the revised manuscript that has already been uploaded, changes and newly added content are shown in blue, and updated figures are highlighted with blue frames.
>
> ## Requested Changes
>
> > (R-1) However, it somehow lacks a remarkable contribution. The authors should perhaps show t-SNE plots of their data sets to be fairer (t-SNE is only reported in quantitative results and tables). Would something like this be a competitor? https://lvdmaaten.github.io/multiplemaps/Multiple_maps_t-SNE/Multiple_maps_t-SNE.html (concatenating X and Y?) or this: https://arxiv.org/abs/2005.00670 ?
>
> Multiple-maps t-SNE and multimodal SNE are related to RFSLVM in that they address multimodal visualization.
> However, they differ in their objectives: RFSLVM is designed to visualize the relationship between items and preference functions and to support downstream analyses through explicit modeling of preference vectors, which is not supported by these embedding-based visualization methods.
> This distinction is reflected in the Introduction (p.2, third paragraph) and in the comparison summarized in Table 1 (p.4).
>
> Following the reviewer’s suggestion, we have additionally included representative t-SNE visualizations in Appendix E (Figures 10 and 11, pp.20–23) to provide a direct visual reference.
>
> To clarify, t-SNE is not reported in the quantitative results or tables of our manuscript.
> The reviewer may be referring to the kNN results reported in Table 4 (p.10).
>
> > (R-2) All in all, there is the feeling that the authors have not challenged their method enough, with other competing methods, although it is true that the framework of multimodality might restrict the list of eligible candidates. Several data sets are assessed quantitatively, whereas only one is illustrated, in a context where visualization is claimed to matter.
>
> To further support the role of visualization, we have added qualitative visualization examples on additional datasets (ANIME and BEER) in the appendix, demonstrating that the qualitative behavior of RFSLVM is consistent across datasets (Appendix E, p.20; Figure 10, p.22; Figure 11, p.23).
>
> Within the limited set of eligible methods, we evaluated both Gaussian process–based and neural network–based multimodal models with stable, publicly available implementations.
>
> ## Weaknesses
>
> > (W-1) On the downsides, the originality of the paper seems to hold mainly in the multimodality and it appears like relatively straightforward extensions of existing methods.
>
> We revised the Abstract, Introduction, and the paper title to clarify that RFSLVM builds upon existing techniques, with the primary contribution being a probabilistic framework that supports multimodal visualization and explicit preference representations for interpretable analysis of item–preference relationships.
>
> > (W-2) Another weak point is that the probabilistic approach and the addition of multimodality seem to have a cost in terms of visualization power. The strong bias towards cluster gap magnification that is typical of more ad hoc methods like t-SNE and UMAP seems to absent here, with cluttered visualizations showing overlapping clusters.
>
> While RFSLVM does not aim to maximize visual cluster separation in the same way as t-SNE or UMAP, it is designed to visualize how items relate to user preference functions, which involves a different trade-off between cluster separation and the visualization of item–preference relationships in the latent space.
>
> As illustrated in Figure 2 (p.3), the visualization focuses on how items are positioned relative to preference intensity contours, supporting the interpretation of item–preference relationships.
> For comparison, we have also added visualizations using t-SNE and UMAP in the appendix (Appendix E, p.20; Figure 10, p.22; Figure 11, p.23) to highlight the differences in visualization behavior.

---

### Decision · Action_Editor_m9BC · 2026-03-23

**Recommendation:** Reject

**Additional Comments:**

The paper was reviewed by three reviewers [Swdp,Qys6,R6br]. Discussion between reviewers and authors was limited, while authors provided rebuttals only one reviewer provided a brief follow-up.

In the reviews, the reviewers appreciated several aspects of the paper:

+ The paper was considered well written and easy to read [Swdp,Qys6], being well organized with clear description of the method [Swdp]
+ The approach was considered well motivated by interpretability needs [R6br]
+ Use of random Fourier features with Gaussian processes was appreciated [Qys6,R6br] with resulting preference vectors seen as enabling downstream analyses [R6br]
+ Simplicity of the generative omdel was appreciated [Qys6]
+ Explicily learning a 2D latent space was appreciated as enabling direct visualization rather than requiring a separate visualization method [R6br]
+ Visualizations were appreciated as a sanity check [Qys6]
+ The experimental plan was considered good [Swdp] and the varied experiments were appreciated as demonstrating interpretability and utility [R6br]

However, several criticisms were also noted by reviewers, and authors provided responses to them.

The originality was criticized:
- The paper was considered a strenghtforward extension of existing approaches [Swdp,R6br] and incremental [Qys6,R6br].
- Originality was mainly sen in the multimodality [Swdp] and incorporation of preference vectors [Qys6], although preference vectors were also seen as standard in kernel approximation an sparse GP literature [R6br].
- Although use of RFFs was appreciated also, it was not seen as a well-known approach [R6br].
Authors made some changes to emphasize building on and integrating existing techniques.

Several other criticisms were also given:
- The reconstruction performance was seen as only slightly/marginally better than MRD and MMVAE [R6br]. Authors argued the result show their latent space supports both reconstruction and preference prediction.
- Technical criticisms of the methodological approach were pointed out [Swdp] in terms of reduced visualization power with the probabilistic approach and added multimodelity [Swdp] and cluttered visualizations of overlapping clusters [Swdp]. Authors responded to the comment claiming different objective of the method.
- Additional discussion of multi-view/multi-modal GPLVM literature was desired [R6br]; audhots added some works.
- Fairness of comparisons was criticized [Swdp] and comparison to t-SNE visualization and alternatives like multiple-maps t-SNE was desired [Swdp]. In a response, authors added t-SNE visualizations.
- Lack of standard recommendation benchmarks or metrics was criticized [Qys6]; authors commented such benchmarks did not natively provide item-level descriptions, and argued their metrics yield insight into how the latent space relates items to user preferences.
- Studying sensitivity to the number of features and random seed was desired [R6br]; authors added runs over random seeds.
- More details were desired for Section 4 [R6br], including discussion of the behavior of GPC; authors added some discussion.
- Additional visualizations of data sets were desired [Swdp]; authors added visualizations of two more data sets.
- Using the visualizations for insights was considered relatively abstract [Qys6]; authors made some changes to help interpret clusters.
- Discussion of scalability was desired [R6br]; authors added mention that extension to larger data sets is future research.
- Discussion of the scope/applicability of the method was desired [Qys6]; authors commented their method is not meant to replace large-scale recommender systems.


Authors provided responses and a resubmitted manuscript. Even after the author responses, a common remaining concern among the reviewers was unconvincing originality [Qys6] and incrementality [Swdp]; in a related note, one reviewer considered that the paper could be interesting to a small audience [R6br]. However, since novelty by itself is not a TMLR criterion, a limited audience may still be sufficient.

Other remaining concerns were about lack of clarity of the core messages partly due to lack of strong use cases/scenarios in Section 5 [Swdp], potential lack of fairness in the comparisons [Swdp], scalability [Swdp], and potential over-interpreting of result significance in abstract and conclusions [Qys6]. These issues can be important for TMLR as they related to convincingness and clarity of the evidence.

After the rebuttals, ultimately, two of the three reviewer recommendations leaned towards acceptance, and one recommendation leaned towards rejection.

Overall, it seems clear that although the novelty is not strong the paper can be interesting as a reasonable combination of existing approaches. However, it is not clear that the writing and the experiments yet are at the level needed for TMLR. It seems that the paper may be close to it, but a further revision and evaluation by the reviewers may be needed to establish it. I view this as something inbetween a minor and major revision; thus, technically I recommend a major revision but I expect it could be done somewhat quickly.

**Audience:**

Yes

**Audience Explanation:**

The paper can be of interest to practitioners who are interested in incorporating multimodality and low-dimensional visualizable latent spaces in modeling of user preferences. Although the originality of the method was criticized, some in the audience could be interested in a method combining the aspects.

**Claims And Evidence:**

No

**Claims Explanation:**

This answer is inbetween a yes and a no.

(See also the similar paragraph in Additional comments)

Even after discussion, the convincingness and clarity of the evidence remained a concern. There are remaining concerns about lack of clarity of the core messages partly due to lack of strong use cases/scenarios in Section 5 [Swdp], potential lack of fairness in the comparisons due to the distinct natures of the methods [Swdp], scalability being an issue for the proposed method and not the competing methods [Swdp], and potential over-interpreting of result significance in abstract and conclusions [Qys6].

It is not clear that addressing these can be done as a minor revision as some of them might be solved as writing issues but some may alternatively need more experiments; the remaining concerns seem to be borderline between a minor and major revision.

**Resubmission Of Major Revision:**

The authors may consider submitting a major revision at a later time.